# Direct observation of autoubiquitination for an integral membrane ubiquitin ligase in ERAD

Basila Moochickal Assainar [1], Kaushik Ragunathan [2] & Ryan D. Baldridge [1,3]

The endoplasmic reticulum associated degradation (ERAD) pathway regulates protein quality control at the endoplasmic reticulum. ERAD of lumenal and membrane proteins requires a conserved E3 ubiquitin ligase, called Hrd1. We do not understand the molecular configurations of Hrd1 that enable auto-ubiquitination and the subsequent retrotranslocation of misfolded protein substrates from the ER to the cytosol. Here, we have established a generalizable, single-molecule platform that enables high-efficiency labeling, stoichiometry determination, and functional assays for any integral membrane protein. Using this approach, we directly count Hrd1 proteins reconstituted into individual proteoliposomes. We report that Hrd1 assembles in different oligomeric configurations with mostly monomers and dimers detected at limiting dilution. By correlating oligomeric states with ubiquitination in vitro, we conclude that Hrd1 monomers are inefficient in autoubiquitination while dimers efficiently assemble polyubiquitin chains. Therefore, our results reveal the minimal composition of a Hrd1 oligomer that is capable of auto-ubiquitination. Our methods are broadly applicable to studying other complex membrane protein functions using reconstituted bilayer systems.

Approximately one-third of all newly synthesized proteins enter the endoplasmic reticulum (ER) where they are folded, undergo quality control, and leave the ER for trafficking towards their final cellular destination[1,2]. Proteins that fail to fold appropriately during this process must be disposed to prevent the accumulation of misfolded proteins in the ER and induction of the unfolded protein response. One way in which ER proteostasis is maintained is through the degradation of misfolded proteins by endoplasmic reticulum-associated degradation (ERAD) pathways, each being centered around different integral membrane E3 ubiquitin ligases[3–5]. The ERAD pathway that is responsible for recognition and degradation of both lumenal and integral membrane proteins depends on the activity of an integral membrane ubiquitin ligase called Hrd1[6,7]. Hrd1-centric ERAD is a multistep process where misfolded protein substrates are recognized, retrotranslocated

from the ER lumen to the cytosol, polyubiquitinated, extracted from the ER membrane, and degraded via the proteasome[2].

Hrd1 can exist in a complex with three other integral membrane proteins called Der1, Usa1, and Hrd3, as well as with a soluble lumenal lectin called Yos9[3,8,9]. Hrd3 assists in substrate selection and controls Hrd1 stability, presumably by controlling Hrd1 autoubiquitination[8,10–13]. Usa1 scaffolds Hrd1/Hrd1 and Hrd1/Der1 interactions along with regulating Hrd1 stability by regulating the activity of a deubiquitinating enzyme called Ubp1[7,10,14]. Under most conditions, Der1 forms half of the retrotranslocon with Hrd1[7,15,16]. Yet, when overexpressed, Hrd1 can degrade ERAD substrates in vivo, even in the absence of any of its interaction partners[7]. In vitro, Hrd1 promotes substrate retrotranslocation in a reconstituted proteoliposome system suggesting a molecular basis for its ability to bypass other ERAD partners[17].

[1]Department of Biological Chemistry, University of Michigan Medical School, 1150 W Medical Center Drive, Ann Arbor, MI 48109, USA. [2]Department of Biology, Brandeis University, 415 South Street, Waltham, MA 02453, USA. [3]Cellular and Molecular Biology Program, University of Michigan Medical School, 1150 W Medical Center Drive, Ann Arbor, MI 48109, USA. ✉e-mail: kaushikr@brandeis.edu; ryanbald@umich.edu

Experiments in this reconstituted system, and in vivo, revealed an essential requirement for Hrd1 autoubiquitination to enable retro-translocation of ERAD substrates, thus suggesting that Hrd1 forms a ubiquitin-gated protein conducting channel[17]. In addition, auto-ubiquitination of Hrd1 triggered conductance of ions across a lipid bilayer supporting the idea that Hrd1 forms an aqueous path for the transport of misfolded proteins from the ER to the cytosolic proteasome[18].

Based on biochemical and structural data, Hrd1 appears to form at least two types of complexes; the Hrd1 transmembrane domain can oligomerize into either a homodimer or can form heterodimers with Der1[15,19,20]. These models may represent different ERAD complex arrangements or distinct complexes for degrading integral membrane versus lumenal proteins. Recent cross-linking experiments suggested that Hrd1 proteins can form homodimers within cells and that Hrd1 autoubiquitination regulates oligomerization by disrupting homo-dimerization in favor of other possible Hrd1-associated ERAD configurations. However, the extent to which the crosslinkers used could influence protein stoichiometry and Hrd1 activity is unclear. In summary, we do not understand how many Hrd1 proteins assemble to form the retrotranslocation channel, whether the ability of Hrd1 to form the retrotranslocon is dynamic, and what the minimal Hrd1 stoichiometry required for autoubiquitination and retrotranslocation is. This is because of a paucity of approaches that can both count the number of Hrd1 proteins in a complex and simultaneously measure whether the complexes are ubiquitination competent.

Here, we develop a single-molecule total internal reflection fluorescence (TIRF) microscopy-based counting platform to define the stoichiometry of Hrd1[21–23]. Using our platform, we directly count the number of Hrd1 proteins reconstituted in a lipid bilayer. With single proteoliposomes containing Hrd1 reconstituted at defined protein to lipid ratios, we determine the minimal stoichiometries capable of autoubiquitination, a basic requirement for retrotranslocation[17,18,20]. We observe that Hrd1 monomers are inefficient in intramolecular autoubiquitination in *cis*, meaning Hrd1 monomers by themselves are unlikely to be functional. Our results suggest that a dimer or higher order oligomer reflects the functional unit of Hrd1 and that a *trans* ubiquitination mechanism involving more than one Hrd1 protein is essential for auto-ubiquitination. Our platform is generalizable and will enable detailed mechanistic studies of other dynamic membrane protein complexes.

## Results

### Development of a high-efficiency single-molecule counting assay

Purified Hrd1 exists as a continuum of dynamic oligomeric species in detergent micelles. To determine the minimal stoichiometry of Hrd1 within a lipid bilayer, we developed a single-molecule photobleaching measurement using Hrd1 reconstituted within individual proteoliposomes. For efficient determination of protein stoichiometry, we needed near-complete fluorescent labeling of Hrd1 to minimize counting artifacts that may arise from the presence of a large excess of unlabeled Hrd1 protein[24]. Previously reported purification and labeling strategies yielded a maximum of 60% Hrd1 labeling[17,25]. We modified the Hrd1 purification and labeling approach by positioning the sortase A recognition motif (Leu-Pro-Glu-Thr-Gly-Gly) in front of a C-terminal streptavidin binding peptide (SBP) epitope tag. During our Hrd1 purification, we eluted Hrd1 with a Cy5 label (Hrd1$^{Cy5}$) from the streptavidin affinity resin by adding recombinant sortase A pentamutant to enzymatically remove the SBP affinity tag and covalently attach a Cy5 coupled peptide[26,27]. We measured the labeling efficiency of our eluted material by visualizing the protein using SDS-PAGE and in-gel fluorescence scanning where we were able to detect a ~5 kDa mobility shift between labeled and unlabeled protein (Fig. 1a). With this modified

procedure, we achieved a Hrd1$^{Cy5}$ labeling efficiency of >95% (Fig. 1a, Supplementary Fig. 1a, b).

We reconstituted Hrd1$^{Cy5}$ into pre-extruded 100 nm liposomes containing DOPC, fluorescently labeled NBD-PC (for liposome visualization), and Biotinyl-Cap-PE (for surface immobilization). Following reconstitution, we floated our proteoliposomes in a glycerol density gradient to ensure that Hrd1$^{Cy5}$ was appropriately integrated into well-sealed bilayers. We observed co-flotation of Hrd1$^{Cy5}$ and NBD-labeled proteoliposomes at the top of glycerol density gradients, demonstrating successful integration of Hrd1$^{Cy5}$ into liposomes (Fig. 1b–d).

Next, we immobilized Hrd1$^{Cy5}$-containing proteoliposomes on a passivated glass coverslip surface functionalized with neutravidin (schematic in Fig. 1e)[28,29]. We visualized Hrd1$^{Cy5}$ NBD proteoliposomes using an objective-type TIRF microscope to detect single proteoliposomes containing reconstituted Hrd1$^{Cy5}$ proteins (Fig. 1f). In the microscope, individual liposomes (~100 nm) appeared as diffraction-limited spots with the fluorescence intensity of each spot corresponding to the number of fluorophores in that particular liposome (Supplementary Fig. 1c). As expected, Hrd1$^{Cy5}$ spots largely colocalized with NBD-PC, consistent with Hrd1 being inserted into the lipid bilayer with unevenly distributed fluorescently labeled lipids even though the liposomes were relatively uniformly sized (Fig. 1f, Supplementary Fig. 1d, e). In addition, we found the majority of Hrd1$^{Cy5}$ was oriented with the cytosolic RING domain facing the outside of the liposomes (Supplementary Fig. 1f).

### Hrd1 assembles into various oligomeric states

Previous functional studies were performed using Hrd1:lipid ratios and reconstitution conditions that would yield approximately 20 Hrd1 proteins per liposome, although it is likely that losses during reconstitution would lead to lower numbers. To determine the appropriate protein to lipid ratios for counting experiments, we titrated different Hrd1:proteoliposome ratios across two orders of magnitude, keeping the concentration of liposomes constant. The reconstitution conditions we tested ranged from 20 Hrd1 per proteoliposome (Hrd1$^{20:1}$) to 1 Hrd1 per 20 liposomes (Hrd1$^{1:20}$, Fig. 2a). For each ratio, we observed similar reconstitution efficiencies, with the majority of Hrd1$^{Cy5}$ and lipid$^{NBD}$ colocalizing near the top of the density gradients in the flotation experiments (Supplementary Fig. 2a). Assuming that the insertion of a membrane protein into a liposome follows Poisson statistics, performing reconstitutions at different ratios of proteins to pre-extruded liposomes would yield binomial distributions, where the probability of successful insertion depends solely on the ratio of proteins to liposomes. This strategy would allow the determination of the optimal ratio at which we can observe enough proteoliposomes for counting while ensuring that the probability for protein co-insertion is negligible[30]. We calculated the theoretical fractions of protein containing liposomes assuming an idealized monomeric or dimeric protein. For the highest ratios of 20 proteins for every liposome (Hrd1$^{20:1}$), almost 99.9% of the liposomes will be occupied regardless of whether the protein is a monomer or dimer. In contrast, at the lowest reconstitution ratios of 1 protein for every 20 liposomes (Hrd1$^{1:20}$), 95% of liposomes would be empty if the protein was a monomer and 97.5% would be empty if the protein was a dimer. The lowest concentration that can be measured represents a practical experimental limit, below which it becomes exceedingly rare to visualize protein-containing liposomes. Experimentally, at our highest Hrd1 concentrations (Hrd1$^{20:1}$), 90% of liposomes were empty whereas with the lowest concentrations (Hrd1$^{1:20}$), >99% of liposomes were empty (Supplementary Fig. 2b).

The Cy5 fluorescence intensity of an individual diffraction-limited foci corresponds to the number of Hrd1$^{Cy5}$ molecules within each diffraction spot although the absolute intensity across the field of view can change due to variations in the TIRF illumination pattern. Photobleaching of individual fluorophores is a discrete event;

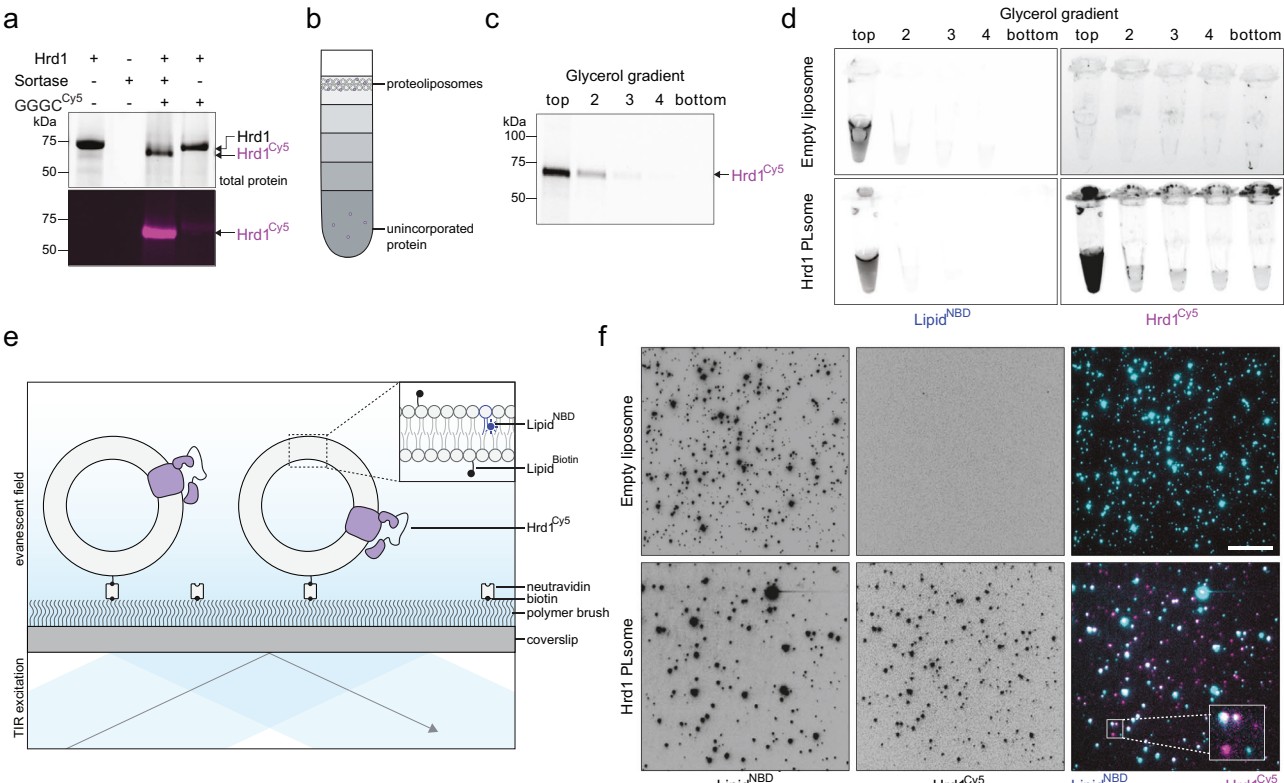

**Fig. 1 | Development of a single-molecule approach to count Hrd1 membrane protein stoichiometry in reconstituted proteoliposomes. a** Purified Hrd1 with a sortase A transpeptidation site and streptavidin binding protein tag (Hrd1-LPETGG-SBP) was immobilized on streptavidin resin and eluted into solution using sortase A and GGGC-Cy5 peptide (GGGC^Cy5) to maximize labeling efficiency. Samples were separated by SDS-PAGE and imaged using Stain free technology to visualize total protein (top panel) and in gel fluorescence for Cy5 (bottom panel). Sortase A elution catalyzes the exchange of a ~5 kDa SBP affinity tag with the GGGC^Cy5 peptide, resulting in >95% of Hrd1 labeled with Cy5 (Hrd1^Cy5). **b** Schematic of the glycerol density step gradient used to float proteoliposomes to verify proteoliposome integrity. Any un-reconstituted protein would be found at the bottom of the density gradient. **c** Hrd1^Cy5 containing proteoliposomes were separated using a glycerol density gradient as in (**b**). The gradient fractions were collected and analyzed as in (**a**). Hrd1^Cy5 was reconstituted at 200 nM in 5 mM total lipid (99% DOPC, 0.5% NBD-PC, 0.5% biotinyl-cap-PE). **d** As in (**c**), but the gradient fractions were analyzed by fluorescence imaging in tubes. For this experiment, Hrd1^Cy5 was reconstituted at 100 nM in 5 mM total lipid. **e** Schematic of Hrd1^Cy5 proteoliposome immobilization using biotinylated lipids on a coverslip surface passivated with a PEG polymer brush. Individual proteoliposomes can be visualized using TIRF excitation. **f** NBD-PC (Lipid^NBD, left panels) and Hrd1^Cy5 (center panels) were visualized using TIRF excitation. Empty liposomes are shown in the top row and Hrd1^Cy5 proteoliposomes are shown in the bottom row. The right panels are overlay images to show colocalization of lipid^NBD (cyan) and Hrd1^Cy5 (magenta). Each diffraction limited spot corresponds to an individual liposome. Hrd1^Cy5 was reconstituted at 100 nM in 5 mM total lipid (N = 3). The inset represents a 3 fold magnified view of the region enclosed within the rectangle and the scale bar is 10 μm. Each panel in this figure is representative of at least three independent biological replicates. Source data are provided as a Source Data file. See also Supplementary Fig. 1.

therefore, the number of photobleaching events in a given time trace should reflect the number of Hrd1 molecules within an individual proteoliposome. This strategy to count photobleaching events gives us information about the stoichiometry of Hrd1 that is independent of the illumination intensity. We extracted the time versus intensity traces from individual diffraction-limited spots and performed manual step counting analysis (Fig. 2b). We categorized proteoliposomes as those containing 1, 2, 3, 4, or those greater than 4 steps (Fig. 2c). Because of our protein labeling efficiency (>95%), we were able to assign the number of Hrd1^Cy5 proteins per proteoliposome with high confidence.

At the highest ratio of Hrd1 to liposome (Hrd1^20:1), ~70% of Hrd1 existed in proteoliposomes with more than 4 photobleaching step per proteoliposome (Fig. 2c). As the ratio Hrd1:liposome in the reconstitution experiments decreased, the population of proteoliposomes with >4 Hrd1s reduced and the populations shifted towards proteoliposomes containing mostly 1 or 2 Hrd1 proteins. At the lowest ratios (Hrd1^1:5 and Hrd1^1:20), monomeric and dimeric Hrd1 populations were associated with the majority of proteoliposomes (Fig. 2c). Even at very low Hrd1:liposome ratios, the relative frequency of Hrd1 dimer barely changed (Hrd1^1:5 with 35% compared to Hrd1^1:20 with 29%), while the monomer became more prominent

(Hrd1^1:5 with 41% compared to Hrd1^1:20 with 61%). These results suggested that we had reached the dilution limit at which point we were observing single insertion events per liposome. Taken together, our results support that Hrd1 is likely to exist in two minimal stoichiometric configurations - monomers and dimers. Hrd1 foci with an undefined stoichiometry (>4) likely arise due to weak interactions between Hrd1 monomer or dimers or multiple discrete insertion events within an individual proteoliposome at high protein concentrations.

A potential limitation of all photobleaching-based analysis methods is whether spots with a single photobleaching event are monomers, rather than an undersampling of dimers[21,22]. Because the protein labeling efficiency is the primary factor that determines the probability of accurately measuring complex stoichiometries, we simulated a binomial distribution of Hrd1 oligomers assuming different protein labeling efficiencies (Supplementary Fig. 2c). We found that at labeling efficiencies of 90% or greater, we could determine protein oligomeric states with high confidence. For a dimeric protein complex labeled with 90% efficiency, the probability of miscounting the complex as a monomer was only 18% (Fig. 2d). With our labeling efficiency of >95%, we could discount an undersampling bias in our data.

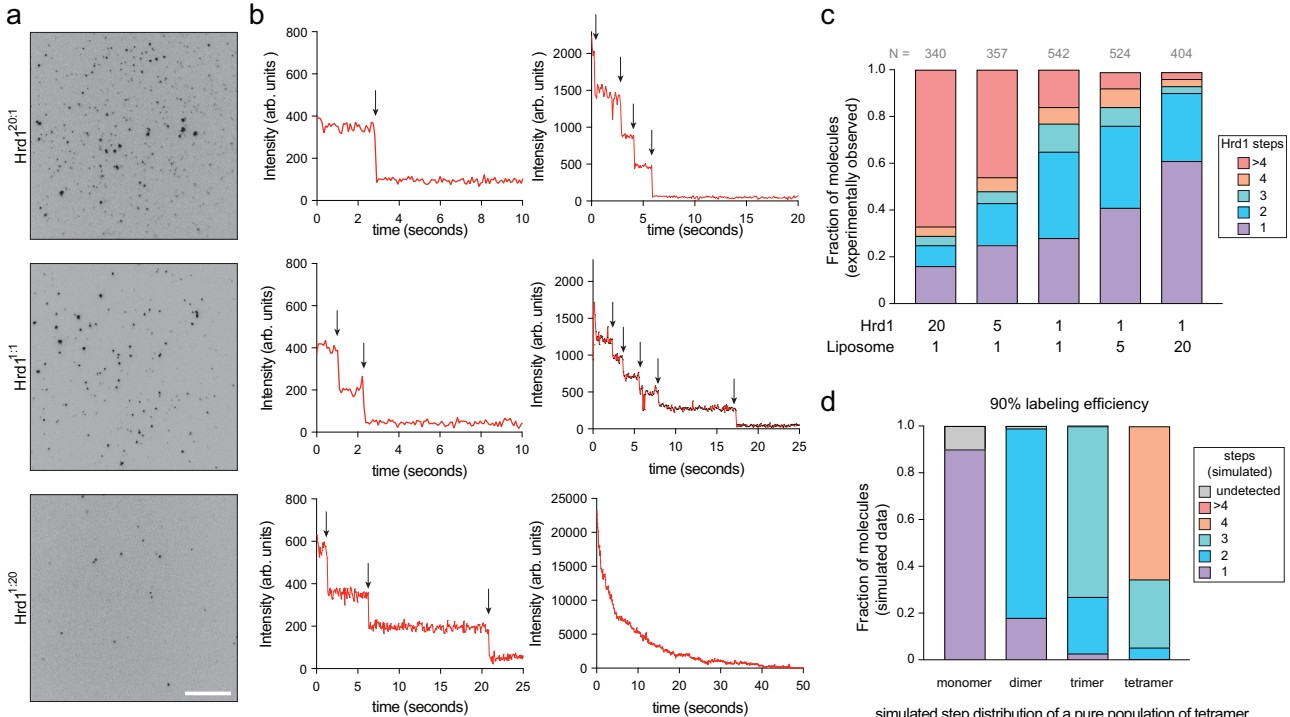

**Fig. 2 | Hrd1 exists in different oligomeric states. a** Proteoliposomes were immobilized on a passivated coverslip surface and Hrd1[Cy5] was visualized under TIRF excitation. Hrd1[Cy5] was reconstituted at different ratios relative to a fixed lipid concentration giving 20 Hrd1 per liposome (Hrd1[20:1]) to 1 Hrd1 per 20 liposomes (Hrd1[1:20]). The scale bar is 10 μm. This panel is representative of at least three independent experiments. **b** Representative photobleaching traces of individual Hrd1[Cy5] proteoliposomes exhibiting 1, 2, 3, 4, 5, or greater than 5 photobleaching events. Black arrows indicate photobleaching steps. **c** Summary of photobleaching step distributions for Hrd1[Cy5] proteoliposomes across five reconstitution ratios. Note that the two lowest concentrations represent the Poisson dilution range for our reconstitution conditions. This panel provides summary data from three independent experiments. **d** Simulated binomial distributions for monomeric, dimeric, trimeric or tetrameric complex with 90% protein labeling efficiency. Source data are provided as a Source Data file. See also Supplementary Fig. 2.

## Reconstitution of ubiquitination at individual protein complexes

Having established a molecular counting assay for Hrd1[Cy5] in proteoliposomes, we turned our attention to testing Hrd1 function in a single-molecule assay. Hrd1 is an E3 ubiquitin ligase and autoubiquitination is necessary to activate Hrd1 mediated retrotranslocation of misfolded ERAD substrates[17]. Since autoubiquitination is a prerequisite for Hrd1 mediated retrotranslocation, we sought to define what types of molecular configurations of Hrd1 were autoubiquitination competent. With the ability to separate Hrd1 into defined oligomeric species, we developed an in vitro single-molecule ubiquitination assay to understand the mechanics of autoubiquitination. We purified ubiquitin and labeled it using Cy3 (ubiquitin[Cy3]). Using our highly loaded Hrd1[Cy5] proteoliposomes (Hrd1[20:1]), we performed in vitro ubiquitination assays with recombinant ubiquitination machinery consisting of ubiquitin[Cy3], Uba1 (E1 enzyme), Ubc7 (E2 enzyme), and Cue1 (E2 accessory protein)[6,25,31–33]. We observed Hrd1 autoubiquitination indicated by the appearance of higher molecular weight products visualized following SDS-PAGE and in gel fluorescence scanning for both Hrd1[Cy5] and ubiquitin[Cy3] (Fig. 3a, Supplementary Fig. 3a). The higher molecular weight products only appeared in the presence of ATP, confirming that the addition of ubiquitin is an energy dependent process and the appropriate function of our system. Next, we immobilized our ubiquitination reactions containing Hrd1[Cy5] proteoliposomes and used TIRF microscopy to determine which Hrd1 oligomers were ubiquitinated (schematic in Fig. 3b). In the absence of ATP, we visualized Hrd1 proteoliposomes but only observed trace amounts of ubiquitin[Cy3], which we attributed to non-specific binding of ubiquitin to Hrd1 containing proteoliposomes, or the passivated glass surface (top row, Fig. 3c). In addition, without ATP the colocalization of Hrd1[Cy5]

with ubiquitin[Cy3] was relatively small with low Cy3 intensity, demonstrating that these signals arise from non-specific surface and liposome interactions, rather than covalent polyubiquitin conjugation events (Fig. 3c). For empty liposomes in the presence of the ubiquitination machinery and ATP, a small fraction of the low intensity ubiquitin[Cy3] colocalized with liposomes (visualized by lipid[NBD], Supplementary Fig. 3b). In the presence of ATP, we observed near-complete colocalization of Hrd1[Cy5] with ubiquitin[Cy3] (Fig. 3c, d). In the presence of ATP, the ubiquitin[Cy3] intensities at foci were significantly higher than in the absence of ATP, likely representing polyubiquitin[Cy3] chains (Fig. 3e). In the highly loaded Hrd1[Cy5] proteoliposomes (Hrd1[20:1]), the majority of proteoliposomes contained more than 4 Hrd1[Cy5] per liposome (Fig. 2c). When we looked at individual Hrd1-containing proteoliposomes, we found that nearly every Hrd1[Cy5] proteoliposome with 2 or more Hrd1[Cy5] was colocalized with ubiquitin[Cy3]. On the other hand, monomers were only colocalized with ubiquitin around 60% of the time (Fig. 3f). The polyubiquitin chain length distributions of individual Hrd1 oligomeric states, showed that larger numbers of Hrd1 correlated with increased ubiquitin[Cy3] intensity (Fig. 3g, Supplementary Fig. 3c).

We anticipated that Hrd1 autoubiquitination using Ubc7/Cue1 should produce K48-linked polyubiquitin chains so we purified a polyubiquitin-specific tandem ubiquitin binding element (TUBE) that is capable of binding polyubiquitin with 6 ubiquitin proteins, and labeled the protein with Cy3 (TUBE[Cy3])[34]. We incubated our Hrd1[Cy5]-containing proteoliposomes with unlabeled ubiquitin and ubiquitination machinery in the absence of ATP, immobilized the proteoliposomes on the surface, and flowed in TUBE[Cy3] (schematic in Fig. 3h). In the absence of ATP, we were unable to observe any colocalization of the TUBE[Cy3] with Hrd1[Cy5] proteoliposomes in our flow cell (Fig. 3i, j). When we performed the same experiment in the presence of ATP, we

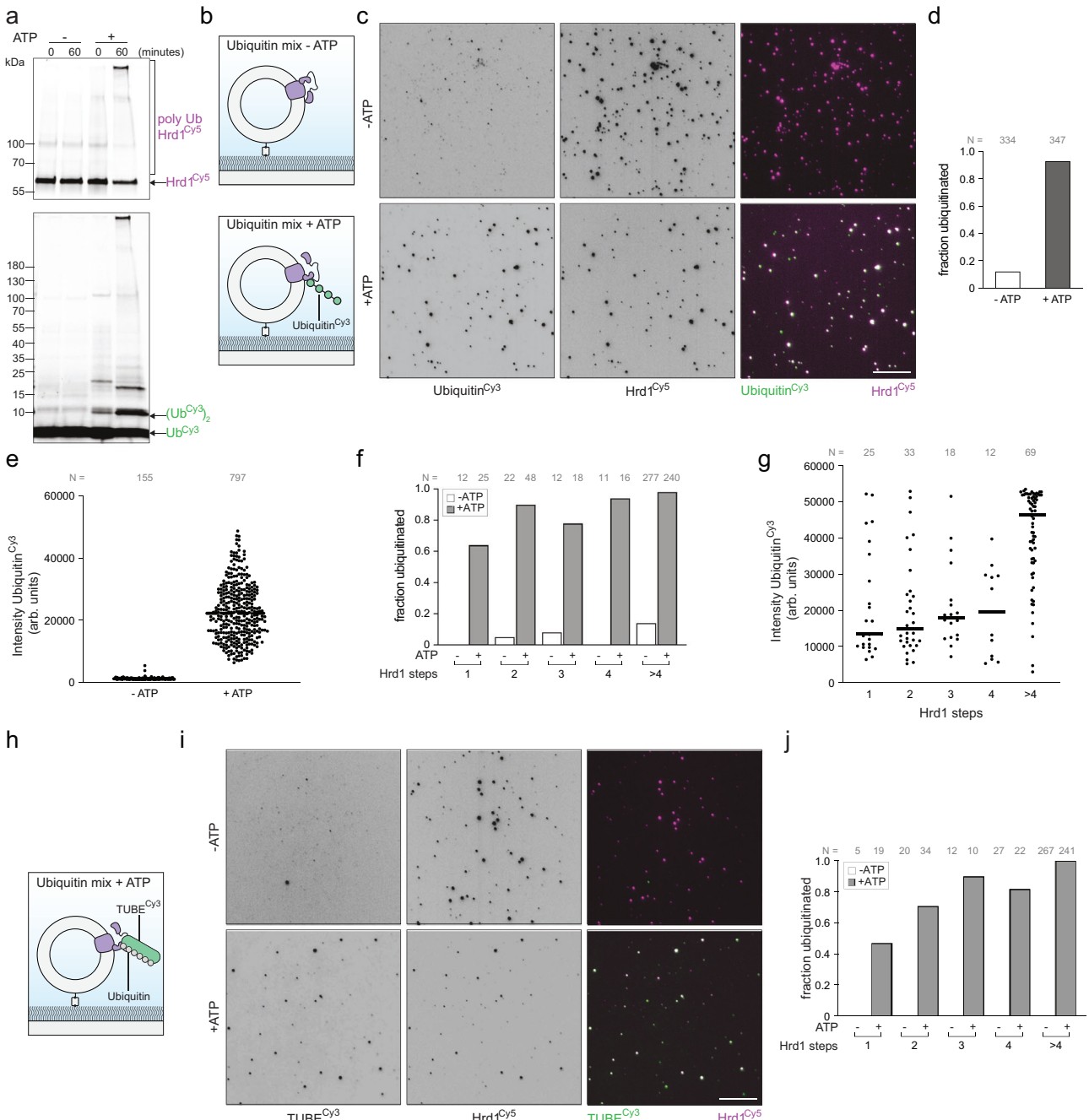

**Fig. 3 | Reconstitution and visualization of Hrd1 autoubiquitination within individual proteoliposomes. a** Hrd1$^{Cy5}$ proteoliposomes were incubated with ubiquitin$^{Cy3}$, recombinant ubiquitin machinery, +/− ATP for the indicated times. The reactions were separated by SDS-PAGE and imaged using in gel fluorescence (for ubiquitin$^{Cy3}$ and Hrd1$^{Cy5}$) or after staining with coomassie blue. **b** Schematic of the ubiquitination experiment with immobilized Hrd1$^{Cy5}$ proteoliposomes. **c** Hrd1$^{Cy5}$ proteoliposomes were immobilized on passivated coverslip surfaces after ubiquitination for 60 min (as in (**a**)) and visualized under TIRF excitation. Ubiquitin$^{Cy3}$ (left panels) and Hrd1$^{Cy5}$ (middle panels) images were overlaid with white showing colocalization (right panels). The scale bar is 10 μm. **d** Quantification of colocalization for ubiquitin$^{Cy3}$ and Hrd1$^{Cy5}$ signals from ubiquitination assay in (**c**). **e** Ubiquitin$^{Cy3}$ intensities for Hrd1$^{Cy5}$ proteoliposomes incubated with ubiquitination machinery +/− ATP. Lines indicate the mean fluorescence intensity. **f** As in (**d**), but separated by Hrd1 oligomeric sizes. For proteoliposomes with a specific number of photobleaching steps, the data is represented as the fraction of colocalized spots to the total number of spots. **g** Ubiquitin$^{Cy3}$ intensity at individual foci separated by

Hrd1 oligomer size in the presence of ubiquitination machinery and ATP in (as in (**c**)). Lines indicate the mean fluorescence intensity. **h** Schematic of unlabeled polyubiquitination detection using a tandem ubiquitin binding element labeled with Cy3 (TUBE$^{Cy3}$) at Hrd1$^{Cy5}$ proteoliposomes. **i** Hrd1$^{Cy5}$ proteoliposomes were immobilized on passivated coverslip surfaces after ubiquitination (using unlabeled ubiquitin) and visualized under TIRF excitation. TUBE$^{Cy3}$ was added to the slide surface at 10 pM and incubated for 1 min before imaging TUBE$^{Cy3}$ (left panels) and Hrd1$^{Cy5}$ (middle panels, overlay in the right panels). The scale bar is 10 μm. j) Quantification of colocalization of TUBE$^{Cy3}$ and Hrd1$^{Cy5}$ signals from the ubiquitination assay in the absence/presence of ATP (in panel (**f**)) separated by proteoliposomes with a specific number of photobleaching steps. Imaging panels in (**a**, **c**, **i**) and the data presented in (**g**) are representative of at least three independent experiments. The summary data in (**d–f**, **j**) combine at least three independent reconstitution experiments. Source data are provided as a Source Data file. See also Supplementary Fig. 3.

observed near complete colocalization of ubiquitinated Hrd1[Cy5]-containing proteoliposomes and the polyubiquitin-dependent binding of TUBE[Cy3] (Fig. 3i, j) with larger Hrd1 oligomers exhibiting increased TUBE binding (Supplementary Fig. 3d). Both methods for detection of Hrd1 autoubiquitination provided similar results, although the directly labeled ubiquitin was much more sensitive (Supplementary Fig. 3e). Taken together, our results demonstrate that we reconstituted Hrd1 autoubiquitination and were able to detect the ubiquitination of an integral membrane protein using both direct (ubiquitin[Cy3]) and indirect approaches (TUBE[Cy3]) with single-molecule sensitivity.

## Hrd1 dimers are required for efficient autoubiquitination

To determine if the time for autoubiquitination depended on Hrd1 stoichiometry we performed a Hrd1[20:1] reconstitution, initiated ubiquitination reactions using ubiquitin[Cy3] and analyzed samples at different times (15 min, 30 min, and 60 min). At the earliest time point (15 min), we observed high levels of ubiquitin[Cy3] labeling of Hrd1 oligomers (>4) but not in the case of other Hrd1 stoichiometries. At later time points, we observed an increase in Hrd1 dimer and trimer populations and low ubiquitination in the case of Hrd1 monomers that exhibited little change (Supplementary Fig. 4a). These results suggested that Hrd1 autoubiquitination is less efficient in the case of monomers but increases as a function of Hrd1 oligomer size, possibly related to avidity effects with the higher-order oligomers being most efficient.

There are at least three ways for Hrd1 molecules to achieve autoubiquitination. First, a Hrd1 monomer could autoubiquitinate itself (intramolecular ubiquitination in *cis*). The second possibility would be for one or more Hrd1 proteins to ubiquitinate each other in an intermolecular reaction within the same proteoliposome bilayer (intermolecular ubiquitination in *trans* between Hrd1 proteins in the same proteoliposome). Finally, Hrd1 proteins within adjacent proteoliposomes might also ubiquitinate each other on opposing membrane surfaces (intermolecular ubiquitination in *trans* but between different proteoliposomes).

To determine whether Hrd1 autoubiquitination was happening in *cis*, *trans*, or in *trans* between different proteoliposomes, we focused on the higher density Hrd1 proteoliposome reconstitutions (Hrd1[20:1]). Here, we clearly observed ubiquitination distributed across both Hrd1 monomers, dimers and higher order oligomers (Fig. 3). We reduced the possibility of ubiquitination reactions in *trans* across different proteoliposomes by adding in a 20-fold excess of empty liposomes to our existing Hrd1[20:1] proteoliposomes (schematic in Fig. 4a). The 20-fold excess of empty liposomes act as spacers to reduce the frequency of interactions between Hrd1 proteins across two independent proteoliposomes, but maintained the same overall concentration of Hrd1 protein in the ubiquitination reaction (Fig. 4a, Supplementary Fig. 4b).

The addition of spacer liposomes had a dramatic effect on the ubiquitination reactions. In contrast to the 60–70% of monomers ubiquitinated in the Hrd1[20:1] reconstitution (Figs. 3f, 4b) addition of spacer liposomes dramatically reduced the fraction of ubiquitinated Hrd1 monomers to 10–20% (Fig. 4b). Additionally, we purified a catalytically inactive Hrd1(C399S), labeled it with Cy5, and reconstituted it at 20:1 ratio in biotin-lipid containing proteoliposomes. We mixed these Hrd1(C399S)[Cy5] proteoliposomes with proteoliposomes containing unlabeled wild-type Hrd1 (also 20:1, but no biotin), performed the ubiquitination assay, and immobilized only the Hrd1(C399S)[Cy5] proteoliposomes on the slide surface. We observed very little colocalization of Hrd1(C399S) with ubiquitin[Cy3] suggesting that *trans* ubiquitination reactions must involve the engagement between two fully functional Hrd1 proteins (Supplementary Fig. 4c, d).

Based on these results, we concluded that the ubiquitination of Hrd1 monomers that we previously observed in our measurements were not due to intramolecular ubiquitination in *cis*. Instead, we interpret these data to mean that efficient Hrd1 autoubiquitination

requires at least two or more Hrd1 proteins to interact in two possible ways: (1) through interactions between two Hrd1 proteins within the same proteoliposome (2) through interactions between two Hrd1 proteins across two proteoliposomes. Taken together, these data demonstrate that a dimer of Hrd1 is the minimal functional unit of autoubiquitination.

## Discussion

Determining the stoichiometry of dynamic integral membrane proteins within a lipid bilayer is a technical challenge. Previous studies have demonstrated counting of membrane protein stoichiometry using either genetically encoded fluorescent proteins or reactive organic fluorophores[21,22,35–40]. Here, we developed a sortase-based labeling strategy that ensures we have highly efficient protein labeling with a single bright, organic fluorophore. Our near-complete labeling strategy (>95% labeling efficiency) eliminates the uncertainty associated with many other single-molecule counting approaches[24]. In addition, we have also developed single-molecule ubiquitination detection assays for an integral membrane protein, and can follow ubiquitination both directly (ubiquitin[Cy3]) and indirectly (TUBE[Cy3]). Importantly, our purification, labeling, and reconstitution strategy can be applied to any integral membrane protein to determine their stoichiometry and function.

Single-molecule approaches have been important to understanding the detailed functional mechanics of the ubiquitin proteasome system[41]. The dynamics of substrate binding, engagement, and processing by the 26S proteasome have been well characterized using real time single-molecule approaches. The most comprehensive single-molecule in vitro ubiquitination studies are those performed using the soluble anaphase-promoting complex (APC)[42,43]. In contrast, single-molecule ubiquitination experiments using integral membrane proteins or integral membrane ubiquitin ligases have not been reported and our platform creates a unique platform to study their mechanics. These methodological innovations for integral membrane E3 ubiquitin ligases are critical since the two-dimensional properties of the membrane constrains the enzyme active site in ways that make the geometry of interaction important. Indeed, our studies reveal that two Hrd1 proteins are absolutely essential for successful ubiquitination reactions.

Our single-molecule platform enabled reconstitution of Hrd1-mediated autoubiquitination. We were able to directly visualize autoubiquitination by specific classes of Hrd1 oligomers using fluorophore-labeled ubiquitin. We found more ubiquitin, and longer polyubiquitin chains, associated with larger Hrd1 oligomers (Fig. 3g, Supplementary Fig. 3c, d). In addition, we were able to validate these results by using an orthogonal method for detection of polyubiquitin chains. We used the polyubiquitin-specific TUBE to identify unmodified polyubiquitin chains directly at Hrd1-containing proteoliposomes (Fig. 3i, j, Supplementary Fig. 3d). Together, these experiments allowed us to assign the relative function of each class of Hrd1 oligomers. One limitation of our approach is that we lack the resolution to confirm whether the inserted complexes continue to interact with each other within a single proteoliposome given their ability to rapidly diffuse in 2D space. However, based on our functional autoubiquitination assays, we believe that Hrd1 within proteoliposomes containing more than one Hrd1 interacts to autoubiquitinate because isolated monomers are inefficient in autoubiquitination (Fig. 4b). At this point, although it is unlikely, we cannot rule out that a Hrd1 monomer is more prone to inactivation during the reconstitution process compared to other forms of Hrd1.

Hrd1 autoubiquitination is absolutely required for retrotranslocation. When Hrd1 is unable to autoubiquitinate at one of three critical residues in its RING domain (Lys373, Lys387, or Lys407) it is completely unable to support degradation or retrotranslocation of lumenal substrates (ERAD-L), either in vivo or in vitro[17,18]. However, a

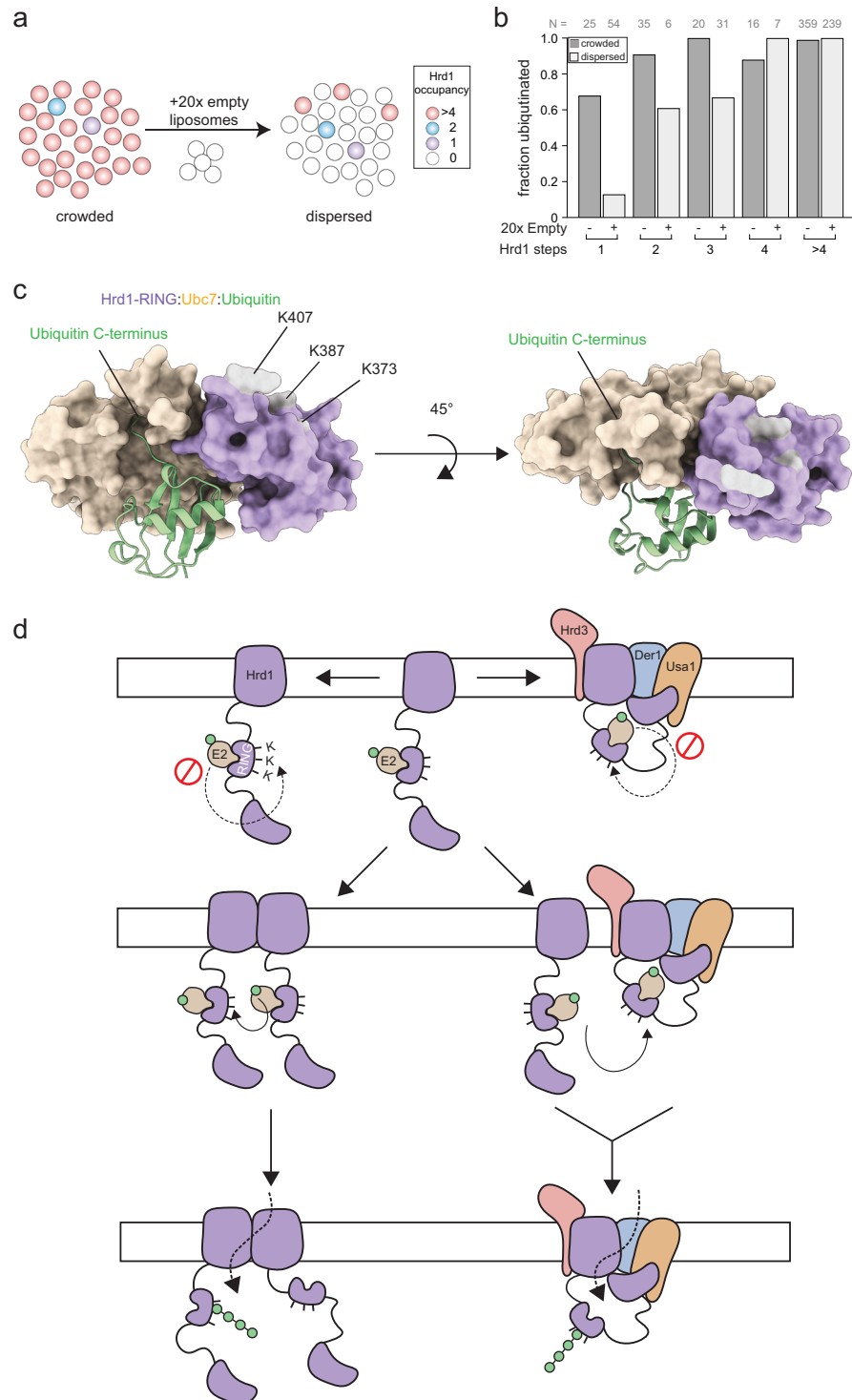

**Fig. 4 | Hrd1 dimers are minimally required for autoubiquitination. a** Schematic of Hrd1[20:1] proteoliposome dilution to reduce the interaction of Hrd1 molecules across different liposomes. **b** Quantification of stepwise colocalization for ubiquitin[Cy3] and Hrd1[Cy5] signals from ubiquitination assay of Hrd1[20:1] diluted 20 fold with empty liposomes. These summary data combine three independent experiments. **c** Alphafold multimer prediction of Ubc7 (yellow), ubiquitin (green), and the Hrd1 RING domain (purple). **d** Model for Hrd1 autoubiquitination. Autoubiquitination requires two Hrd1 proteins and cannot happen in *cis*, but only in *trans*. Each imaging panel is representative of at least three independent experiments and summary data combine three independent experiments. Source data are provided as a Source Data file. See also Supplementary Fig. 4.

Hrd1 molecule that is unable to achieve autoubiquitination within its RING domain appears to be partially functional for the degradation of integral membrane protein substrates (ERAD-M)[10]. Based on the structural prediction of RING domain-E2 interactions[44], we predicted that it was unlikely that ubiquitination of Hrd1 Lys373, Lys387, or

Lys407 is possible through an E2 interacting with a Hrd1 RING finger intramolecularly in *cis* (Fig. 4c). In fact, our results clearly demonstrate that a Hrd1 monomer is unable to autoubiquitinate intramolecularly in *cis*, and thus overall supports our hypothesis of the RING-E2 structural configuration (Fig. 4b).

This is likely to be the case even when Hrd1 heterodimerizes with Der1. We predict a sequential model of transport wherein Hrd1 autoubiquitination first occurs through an intermolecular reaction in *trans* (Fig. 4d). The ubiquitinated Hrd1 protein can form a heterotetrameric complex of Hrd3, Hrd1, Der1, and Usa1 that is proposed to function as the retrotranslocon[15]. Activation and inactivation would require either disassembly of the complex and reassembly following ubiquitination, or a second Hrd1 (or Hrd1 complex) to ubiquitinate and activate the assembled Hrd1 complex directly. This model is consistent with a recent study that suggested Hrd1 autoubiquitination dissociates Hrd1 homodimers to enable Hrd1 interaction with Der1[20]. In addition, Usa1 is known to bridge interactions between multiple Hrd1 proteins, which is required for efficient ERAD indicating that this type of trans-complex interaction is normally important, although not essential[7,15].

It is noteworthy that in vitro reconstituted systems still required autoubiquitination for retrotranslocation, even when Der1 was absent[17,18,25]. In fact, autoubiquitination generates a new substrate binding site on the cytosolic face of Hrd1 that is important for retrotranslocation[18,27]. We favor a model in which Hrd1 autoubiquitination generates a conformational change that both promotes the disassembly of larger oligomers into functional units (heterodimers with Der1 or homodimers of Hrd1), while concurrently exposing a substrate binding site that drives the initial retrotranslocation step. Based on the different complexes identified biochemically, structurally, and the heterogenous oligomers we reconstituted in proteoliposomes, it is likely that Hrd1 also exists in different configurations in the cell[7,15,17,19,20,25,45]. We suggest that each of these complexes are parts of an important regulatory cycle wherein ERAD selectively degrades different types of substrates.

Our experiments also revealed ubiquitination reactions that can occur between independent proteoliposomes. This interproteoliposome event could indicate a new mode of function for Hrd1. Given the intricate ER tubular network (in mammalian cells) or even fenestrations in ER sheets[46], Hrd1 proteins could reach across ER tubules to activate each other rather than engaging with neighboring Hrd1 molecules in close proximity within the same lipid bilayer. However, the physiological significance of such an interaction, or possibility within a cell, is unclear and requires additional investigation. Currently, we cannot exclude the possibility that the inter-proteoliposome ubiquitination reaction is an artifact of our in vitro reconstitution system where Hrd1 proteoliposomes are present at high densities.

In this study, we developed a single-molecule approach to study the stoichiometry and function of a dynamic integral membrane protein. Our findings revealed that Hrd1 forms different classes of oligomeric species in a lipid bilayer. At the lower limit of our reconstitution assays, we detected monomers and dimers. These results suggest that Hrd1 can exist as either a monomer or dimer in a lipid bilayer, which might explain how a Hrd1 monomer can assemble in heteromeric complexes with other proteins such as Der1, Hrd3, and Usa1 to enable different functions in the context of ERAD. In summary, using a single-molecule fluorescent counting approach, paired with single-molecule ubiquitination, we defined minimal and functional complexes of the highly-conserved E3 ligase Hrd1. We anticipate that our platform will serve as the basis for future mechanistic studies in ERAD, other integral membrane ubiquitin ligases, and diverse types of integral membrane proteins.

## Methods
### Strains and plasmids
Yeast deletion strains used in this study were derivatives of BY471 and BY4742. The Hrd1 expression strain was a diploid *hrd1Δubc7Δ* strain (yBGP55B: *MAT* A/α *his3Δ1/his3Δ1 leu2Δ0/leu2Δ0 LYS2/lys2Δ0 met15Δ0/MET15 ura3Δ0/ura3Δ0 hrd1::HphNT1/hrd1::HphNT1 ubc7::KanRMX4/ubc7:KanRMX4*) generated by crossing *hrd1Δubc7Δ*

mat A with *hrd1Δubc7Δ* mat α cells. Yeast transformations were performed using the lithium acetate method[47]. Bacterial protein expression was from BL21-CodonPlus (DE3)-RIPL cells (Agilent Technologies), unless otherwise stated. Plasmids were generated using standard restriction cloning or Gibson assembly. For a list of plasmids used in this study see Supplementary Table 1.

### Protein purification and labeling
Sortase A, Uba1, Ubc7, and Cue1 were purified as described previously[25–27].

**Ubiquitin expression and purification.** Ubiquitin was expressed containing an N-terminal cysteine replacing the initiator methionine with an N-terminal His$_{14}$-SUMO fusion tag in *E. coli*.[48] Cells were inoculated in terrific broth (2.4% yeast extract (w/v), 2.0% tryptone (w/v), 17 mM KH$_2$PO$_4$, 72 mM K$_2$HPO$_4$, 0.4% glycerol (v/v)) at ~0.1 OD$_{600}$/ml and grown at 37$^0$C with shaking to 1.0 OD$_{600}$/ml. The temperature was shifted to 18 °C and protein expression was induced with 0.25 mM Isopropyl β-d-1-thiogalactopyranoside (IPTG) and grown with shaking for 16 h. The cells were pelleted and washed once with water before resuspending in lysis buffer (20 mM Tris (pH 8.0), 500 mM NaCl, 20 mM imidazole, freshly added 1 mM phenylmethylsulfonyl fluoride (PMSF), and protease inhibitor cocktail (100 µM AEBSF, 0.6 µM Aprotinin, 1 µM E-64, 10 µM Leupeptin, 5 µM Pepstatin A, 5 µM Bestatin) and lysed by sonication. The crude cell lysate was clarified by ultracentrifugation in a Type 45 Ti rotor at 147,500*g* for 33 min at 4 °C. The clarified supernatant was rolled at 4 °C with HisPur Ni-NTA Resin (1 mL resin per liter of cells, Thermo Scientific) for 2 h. The resin was washed with 25 CV of lysis buffer and eluted with the lysis buffer supplemented with 400 mM imidazole. The elutions were supplemented with 10% glycerol and 0.5 mM tris(2-carboxyethyl)phosphine (TCEP) before rolling with 30 µM of purified Ulp1 (SUMO protease) overnight at 4 °C, while dialysing against the lysis buffer with 1 mM TCEP and 10% glycerol. The His$_{14}$-SUMO tag was separated from cys-ubiquitin by passing the dialyzed eluate over HisPur Ni-NTA Resin (Thermo Scientific). For labeling, the aliquot with cys-ubiquitin was degassed and incubated with a 2-fold molar excess of Sulfo-Cyanine3 maleimide (Lumiprobe) dissolved in DMSO for 2 h on ice. After labeling, the reaction was passed over a Sephadex G25 column in gel filtration buffer (10 mM HEPES (pH 7.4) and 100 mM KCl) for buffer exchange and to separate the uncoupled Cy3 from ubiquitin$^{Cy3}$. The fractions were analyzed by SDS-PAGE, concentrated to 2.2 mg/ml, flash frozen in liquid nitrogen, and stored at −80 °C.

**TUBE expression and purification.** 6x trypsin resistant tandem ubiquitin binding element (6x TR-TUBE) with an N-terminal hexahistidine tag and T7 tag in pRSET plasmid was a gift from Yasushi Saeki (Addgene plasmid # 110313)[49]. Cells were inoculated in terrific broth at ~0.1 OD$_{600}$/ml and grown at 37 °C with shaking to 1.0 OD$_{600}$/ml. The temperature was shifted to 18 °C and protein expression was induced with 0.1 mM IPTG and grown with shaking for 16 h. The cells were pelleted and washed once with water before resuspending in lysis buffer (20 mM Tris (pH 8.0), 500 mM NaCl, 20 mM imidazole, freshly added 1 mM phenylmethylsulfonyl fluoride (PMSF), and protease inhibitor cocktail ((100 µM AEBSF, 0.6 µM Aprotinin, 1 µM E-64, 10 µM Leupeptin, 5 µM Pepstatin A, 5 µM Bestatin) and lysed by sonication. The crude cell lysate was clarified by ultracentrifugation in a Type 45 Ti (45Ti) rotor at 147,500*g* for 33 min at 4 °C. The clarified supernatant was rolled at 4 °C with HisPur Ni-NTA Resin (1 mL resin per liter of cells, Thermo Scientific) for 2 h. The resin was washed with 25 CV of lysis buffer, degassed, purged with nitrogen gas and rolled with 2 CV of 160 µM sulfo-Cyanine3 maleimide (Lumiprobe) at 4 °C overnight. The following day, the resin was washed with 2 CV of lysis buffer to remove any free or unreacted dye. Another 2 bed volume of degassed 160 µM sulfo-Cy3 maleimide dye was added and rolled for an additional 30 min

at 4 °C. After labeling, the resin was washed with 25 CV of lysis buffer and TUBE[Cy3] was eluted from the resin with lysis buffer supplemented to 400 mM imidazole. The eluted protein was dialyzed against 20 mM Tris (pH 8.0), 50 mM NaCl and loaded onto HiTrap Q XL column (Cytiva) pre-equilibrated with the same buffer. The protein was eluted with 20 mM Tris (pH 8.0) and a linear gradient from 50 mM NaCl to 500 mM NaCl over 100 CV with the protein eluting around 250 mM NaCl. Peak fractions were analyzed by SDS-PAGE for purity and were pooled, before concentration and gel filtration using on a Superdex 75 Increase 10/300 GL column equilibrated with 50 mM HEPES (pH 7.4) 300 mM KCl, 0.5 mM TCEP. The fractions were analyzed by SDS-PAGE, concentrated to 15 mg/ml, flash frozen in liquid nitrogen, and stored at −80 °C.

**GGGC maleimide coupling.** A Gly-Gly-Gly-Cys peptide with C-terminal amidation (Genscript) was dissolved at 173 mM in degassed coupling buffer (50 mM HEPES (pH 7.4), 20 mM TCEP (pH adjusted to 7.5)). Sulfo-Cyanine5 maleimide (Lumiprobe) dissolved in degassed DMSO was mixed in a 2:1 molar ratio (dye:peptide) and incubated overnight at 4 °C. The following morning, the reaction was quenched with 10 mM DTT. The labeling reaction mixture was separated using on a C-18 reverse phase HPLC Column (Beckman Ultrasphere C-18, 4.6 25 cm) that was pre-equilibrated with Buffer 1(filtered MilliQ with 0.1% Trifluoroacetic acid (Sigma-Aldrich)). The protein was eluted at 1.5 ml/min using the following gradient- 0–1 min: 10% buffer 2 (filtered Acetonitrile (Sigma-Aldrich) with 0.1% TFA); 1–16 min: 70% buffer 2; 16–17 min: 90% buffer 2; 17–22 min: 90% buffer 2. The column eluent was monitored at 200 nm and 650 nm. Under these conditions, unlabelled GGGC peptide eluted at ~11 min and GGGC[Cy5] peptide eluted at ~15 min, respectively, under these conditions. The free Cy5 dye was eluted from the column only after washing with 100% buffer 2. The GGGC[Cy5] peptide was lyophilized and stored at −80 °C.

**Hrd1 expression and purification.** Yeast cells containing a 2-micron plasmid with GAL1-driven Hrd1 or Hrd1 C399S with a C-terminal sortase A recognition sequence and streptavidin binding peptide (SBP) (with the C-terminal tag being GSLPETGGGGLEVLFQGPGSGM DEKTTGWRGGHVVEGLAGELEQLRARLEHHPQGQREP) were grown in synthetic dropout media supplemented with the required amino acids (SD + 2% glucose). Starter cultures were grown until mid-log phase shaking at 30 °C and used to inoculate larger expression cultures (approximately 1:150 dilution). Expression cultures were grown for 24 h at 30 °C with shaking to deplete glucose before induction of Hrd1 expression by addition of 2% galactose. The cultures were shifted to 25 °C and shaken for 17 h. Cells were harvested by centrifugation, resuspended in 2 mM DTT for 15 min on ice (to weaken the cell wall for lysis), pelleted and stored at −80 °C.

Cells were resuspended in buffer A (50 mM HEPES (pH 7.4), 300 mM KCl, 0.5 mM TCEP) supplemented with fresh 1 mM PMSF and 1.5 µM Pepstatin A. The cells were lysed with 0.5 mm borosilicate beads in a BioSpec Bead Beater for 30 min with 25% duty cycle (15 s on, 45 s off). After lysis, all steps were performed at 4 °C, unless otherwise noted. The lysate was decanted away from the glass beads and centrifuged at 2000*g* for 10 min to clear unbroken cells and nuclei. The resulting supernatant was subjected to a second spin (2000*g*, 10 min). The supernatant was centrifuged in a 45Ti rotor at 147,500*g* for 33 min. The membrane pellet was resuspended in buffer A with fresh PMSF and Pepstatin A using a loose-fitting Dounce homogenizer. The ultra-centrifugation and resuspension was repeated twice. The final membrane fraction from the third ultracentrifugation was resuspended in buffer A with 1% decyl maltose neopentyl glycol (DMNG) and rolled for 60 min. The detergent-solubilized membrane fraction was centrifuged in a 45Ti rotor at 147,500*g* for 33 min to remove the insoluble material. The detergent-solubilized supernatant was rolled overnight with high

binding-capacity streptavidin agarose resin (Pierce). The streptavidin resin was subsequently washed with 25 CV buffer A (+1% DMNG), 25 CV buffer A (+1 mM DMNG), 25 CV buffer A (+120 µM DMNG), 25 CV buffer A (+120 µM DMNG, 0.5 mM ATP, at room temperature), and 3 × 25 CV buffer A (+120 µM DMNG). Hrd1 was eluted and labeled by rolling for 30 min with 2 CV of buffer A substituted with 120 µM DMNG, 10 mM CaCl₂, 0.5 mM GGGC[Cy5], 5 µM sortase A (pentamutant)[50]. The resin was washed with 1 CV buffer A (+120 µM DMNG). The elution and washing step was repeated three additional times. The elution and wash fractions containing Hrd1[Cy5] were analyzed using SDS-PAGE, pooled, and concentrated before gel filtration using a Superose 6 Increase 10/300 GL column equilibrated in buffer A (+120 µM DMNG). The fractions were analyzed using SDS-PAGE, pooled, concentrated to 2 mg/ml, flash frozen in liquid N₂, and stored at −80 °C.

### Reconstitution into liposomes

Lipids solubilized in chloroform were mixed at 99% DOPC (1,2-dio-leoyl-sn-glycero-3- phosphocholine (Avanti Polar Lipids, Inc.)), 0.5% NBD-PC (1-palmitoyl-2-(6-[(7-nitro-2-1,3- benzoxadiazol-4-yl)amino] hexanoyl)-sn-glycero-3-phosphocholine (Avanti Polar Lipids, Inc.)), and 0.5% biotin-PE (1,2-dioleoyl-sn-glycero-3-phosphoethanolamine-N-(cap biotinyl)(sodium salt)(Avanti Polar Lipids, Inc.))(by mass). Chloroform was evaporated under a stream of N₂ gas to produce a lipid film and trace solvent was removed by lyophilization overnight. The lipid film was resuspended in buffer U (50 mM HEPES (pH 7.4), 150 mM KCl, 2.5 mM MgCl₂, 0.5 mM TCEP) by vortexing at room temperature to form liposomes. Liposomes were extruded using a Mini Extruder and polycarbonate membranes with 100 nm pores (Avanti Polar Lipids) to form uniformly sized unilamellar liposomes. The liposomes were partially solubilized with 0.1% DMNG at room temperature for 30 min. The liposomes were cooled on ice and Hrd1[Cy5] was added at the concentration indicated in each experiment, and incubated for 60 min. To remove detergent, the reconstitution mixture was incubated with detergent removal resin (Pierce) at a ratio of 3.5 volumes resin per volume of liposome solution for 30 min on ice, and eluted by centrifugation. The detergent removal step was repeated three more times. The reconstituted liposomes were mixed 1:1 with 80% glycerol in buffer U and layered under a manually assembled glycerol step gradient of 40%, 30%, 15% and 0% glycerol buffer U. The step gradient was centrifuged in a TLS-55 rotor at 166,000*g* at 4 °C for 3 h. After centrifugation, the gradient was carefully disassembled into 5 layers starting from the top. Under the conditions used in this study, the proteoliposomes floated to near the top of the gradient. To determine the orientation of Hrd1, proteoliposomes were incubated with 0.1 mM unlabelled GGGC peptide and 1 µM sortase A (pentamu-tant) for 2 h on ice, in the presence or absence of 10 mM Ca²⁺ and/ or 1% DMNG.

We added Hrd1 at five different concentrations (between 2 µM and 4 nM) into a fixed concentration of liposomes to do the counting experiments. For a liposome of diameter 50 nm with Hrd1 added at 2 µM, the ratio of number of Hrd1 to the number of liposomes is approximately 8. Similarly, for a liposome of diameter 100 nm, the ratio will be approximately 35. Given we extruded our liposomes, we expect a heterogeneous population primarily composed of liposomes of radius 100 nm, alongside a distribution containing smaller radii. We measured the size of the reconstituted proteoliposomes using dynamic light scattering at 20 °C (Uncle Instrument, Unchained Labs). Given this, we assumed the ratio of the number of proteins to the liposome to be 20.

### Ubiquitination assays

In vitro ubiquitination of Hrd1[Cy5] proteoliposomes was performed as described previously[17,25]. The ubiquitination reaction consisted of 0.4 µM Uba1, 4 µM Ubc7, 4 µM Cue1ΔTM, 0.6 µM BSA (Bovine Serum Albumin), and either 10 µM yeast ubiquitin (R&D Systems) or 10 µM

ubiquitin[Cy3], except for experiments summarized in Supplementary Fig. 3E where 10% of the total ubiquitin was labelled with Cy3 (1 μM ubiquitin[Cy3], 9 μM unlabelled ubiquitin). Proteoliposomes comprised approximately 1/5th of the final volume of ubiquitination reaction. Proteoliposomes with the ubiquitin reaction components were brought to room temperature for 5 min and the reaction was initiated by the addition of 2 mM ATP, or a buffer control. Ubiquitination reactions were incubated at 30 °C for 15, 30, or 60 min. Samples from the in vitro ubiquitination were used in the microfluidic channels and/ or analyzed by SDS-PAGE with in gel fluorescence scanning and coomassie blue staining.

### Single-molecule imaging

**Glass surface-passivation and microfluidic chamber assembly.** All single-molecule experiments were performed in microfluidic chambers assembled in-house. A microfluidic chamber is made of surface treated glass slide and coverslip with inlet and outlet for sample application. To make the chambers 4 pairs of holes were drilled through a glass slide (Thermo Scientific 25*75 mm, 1 mm thickness), which served as the inlet and outlet for sample injection. Slides and glass coverslips (VWR, 24*40 mm) were placed in a glass staining jar for cleaning, cleaned by bath sonication in MilliQ for 5 min, then sonicated in acetone for 30 min. Slides and coverslips were etched by treatment with 3 M potassium hydroxide between 2 and 3 h, with the first 40 min in a bath sonicator. The etched glass surfaces were functionalized using an amino silanization mixture (5:3:100 of glacial acetic acid:(3-Aminopropyl)trimethoxysilane:methanol). Two rounds of pegylation were performed to ensure complete passivation. For the first round of PEGylation, 70 μl of PEG mixture was added to the slide (2.2 mg Biotin-PEG-SVA, MW 5000 (Laysan Bio Inc., Part # Biotin-PEG-SVA-5000-100mg) and 6 mg mPEG-SVA, MW 5000 (Laysan Bio Inc., Part # MPEG-MAL-5000-1g) in 64 μl 0.1 M NaHCO₃) and an etched coverslip was placed carefully on top, without trapping any air bubbles, and left to react overnight in a dark humid environment. On the following day, the slide and coverslip were washed with MilliQ water. The PEGylation was repeated, but with incubation for 2 h. The PEGylated slides and coverslips were washed, dried and stored under vacuum at −20 °C. Microfluidic channels were assembled by placing double-sided tape on the slide surface between the pairs of drilled holes and affixing a coverslip carefully on top. The edges of the channels were sealed with epoxy.

**Immobilization of proteoliposomes on the slide.** The liposomes were immobilized on the glass surface via biotin PEG-neutravidin-biotin lipid interactions. For this, the assembled microfluidic channels were washed with buffer U, and incubated with 0.5 mg/ml of Neutravidin (ThermoScientific) for 5 min. The channels were washed with buffer U and the liposomes were added into the channel (started with 1000 fold dilution of Hrd1[20:1] proteoliposome) and incubated for 15 min to allow immobilization. The unbound liposomes were washed away with buffer U and the coverslip surface was imaged as described below. For orthogonal detection of polyubiquitination of Hrd1 proteolipsomes, 10 pM TUBE[Cy3] was added to surface immobilized preubiquitinated proteoliposomes and incubated for 1 min before imaging.

**Image acquisition.** All single-molecule experiments were recorded on a Nikon Eclipse Ti inverted objective microscope with a Nikon 100 × 1.45 numerical aperture oil-immersion objective, a TIRF illuminator, Nikon Perfect Focusing system, and a motorized stage. The samples were illuminated using 532 nm diode laser or 638 nm solid state laser controlled using a commercially available Oxxius Laser combiner. The Di03-R405/488/532/635-t1 quad filter cube was used in combination with long pass emission filters mCherry (while imaging at 532 nm Laser)/Cy5 (while imaging at 638 nm excitation) controlled

using Lambda 10-B SmartShutter controller. Movies were recorded on a cooled ANDOR iXon Life EMCCD camera which was connected downstream to a Cairns Optosplit II module for dual color imaging. All movies were acquired using NIKON NIS elements software.

For dual color experiments, proteoliposomes were first imaged using a 638 nm laser excitation to follow Hrd1[Cy5] until the observed area had photobleached. Then, the same area was imaged either using 532 nm or 488 nm excitation to image Cy3 labeled molecules or NBD labeled lipids. The fraction of molecules that colocalized between Hrd1[Cy5] and Cy3 labeled ubiquitin[Cy3] were counted by mapping the corresponding position of spots in each channel. Single-molecule colocalization experiments for Figs. 3, 4 were performed in ubiquitination buffer without any photostabilizers to facilitate faster photo bleaching for step counting analysis

For the experiments in Fig. 2, single-molecule counting assays were performed in buffer U supplemented with 0.4% (w/v) glucose, 10 mM (±)−6-Hydroxy-2,5,7,8-tetramethylchromane-2-carboxylic acid (Aldrich), 1 mg/ml Glucose-coupled Glucose Oxidase (Sigma), 0.04 mg/ml Catalase (Sigma).

**Image analysis.** All image analysis was performed using Nikon NIS-Elements AR Analysis 5.02.01. The spot detection tool was used to detect proteoliposomes and generate time versus intensity traces.

The criteria used for exclusion of spots from step analysis were: drifting during imaging, non-spherical in shape which indicates improper focus, and spot crowding leading to ambiguity in picking single proteoliposomes. These plots were analyzed manually and categorized into those with 1, 2, 3, 4, or >4 steps. Traces that had an unclear number of steps were discarded. For colocalization experiments with ubiquitin[Cy3], we used NIS- Elements to directly map the colocalized spots in both channels.

**Simulation of step distribution using binomial distribution.** For fluorescence based molecular counting assays, poor labeling efficiency of the protein could result in undercounting the number of subunits forming a complex[21]. The theoretical probability distribution of subunit distribution can be described using the following equation:

$$P_{n,k} = \frac{n}{k} p^k (1-p)^{(n-k)} \tag{1}$$

where p is the fluorescent labeling efficiency of the protein

**Simulation of membrane protein capture within liposome membrane using Poisson statistics.** Fraction of liposomes with n protein is given by[30],

$$f_n = \frac{\lambda^n e^{-\lambda}}{n}! \tag{2}$$

$$\lambda = \text{Number of protein/number of liposome}$$

**Hrd1-Ubc7-Ubiquitin complex structure prediction.** We used COSMIC[2], a freely available cloud platform[51], to run Alfafold Multimer (2.3.2)[44], to predict protein-protein complex of Hrd1 RING domain, Ubc7 (E2) and Ubiquitin.

### Reporting summary

Further information on research design is available in the Nature Portfolio Reporting Summary linked to this article.

## Data availability

All data required to interpret the results in this paper are provided within the main text, supplementary material, and source data. Any additional data in this study are available from the corresponding authors upon request. Source data are provided with this paper.

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

## Acknowledgements

The authors would like to thank Amanda Ames, Melissa Seman, Jiwon Hwang and Jeremy Dortch for their critical reading of the manuscript and the members of the Baldridge, Ragunathan and Mosalaganti labs for their thoughtful discussion and comments regarding this work. We thank Markus Ruetz for helping us with GGGC$^{Cy5}$ peptide purification. This work was supported by the University of Michigan Medical School Biological Sciences Scholars Program and a NIH/NIGMS Award (R35GM128592) to R.D.B, NIH/NIGMS Award (R35GM137832) to K.R, and a Rackham precandidate fellowship to B.M.A.

## Author contributions

B.M.A. conceptualized the project, developed the methodology, performed the investigation, analyzed the data, visualized the data, wrote the original draft, and reviewed/edited the final manuscript. R.D.B. conceptualized the project, performed the investigation, analyzed the data, visualized the data, wrote the original draft, reviewed/edited the final manuscript, and supervised the study. K.R. conceptualized the project, performed the investigation, analyzed the data, visualized the data, wrote the original draft, reviewed/edited the final manuscript, and supervised the study.

## Competing interests

The authors declare no competing interests.
