## [Peer Review File · Nature Communications]

Direct observation of autoubiquitination for an integral membrane ubiquitin ligase in ERADREVIEWER COMMENTS

Reviewer #1 (Remarks to the Author):

In the manuscript "Direct observation of autoubiquitination for an integral membrane ubiquitin ligase in ERAD" by Assainar et al., the authors investigate the minimal multimerization state required for the ubiquitin ligase Hrd1 to be active in the sense that it causes its autoubiquitination. They develop a single-molecule imaging assay based on proteoliposomes. After purification and labeling of Hrd1 with Cy5, the protein gets integrated into the liposomes. Importantly, highly efficient labeling (>95%) was achieved with a newly developed sortase-based approach. The number of Hrd1 molecules integrated into individual liposomes was determined with a subunit-counting method based on Cy5 photobleaching steps. For low amounts of protein, the authors see mainly one or two proteins per proteoliposome. Next, they used Cy3-labeled ubiquitin to test if the proteins are able to induce autoubiquitination. First, they observe that a majority of all proteoliposomes get ubiquitinated. However, when they add empty liposomes as spacers, the proteoliposomes displaying a single photobleaching step do not show ubiquitination, but all the others still do. Their conclusion is that at least two Hrd1 molecules in the same liposome are required for autoubiquitination.

I think that the study is very well done and shows a novel combination of approaches for single-molecule analysis. The observation that the proteoliposomes with a single photobleaching step do not show ubiquitination after dilution with the spacer liposomes is very exciting. However, at this point, the authors fail to test their two hypotheses that the underlying cause is indeed that monomers are unable to lead to autoubiquitination, and that the lack of spacer liposomes allow trans-ubiquitination of the monomers by active complexes in other proteoliposomes. I am quite surprised that the authors did not do these experiments since (in my view) no additional tools or developments would be needed. Therefore, in the current state, I would consider this work as somehow incomplete and recommend a revision of the manuscript to allow addition of the experiments as suggested in major comment 4: (i) determining the fraction of ubiquitinated proteoliposomes in the 1:20 dilution, where proteoliposomes contain only 1 or 2 Hrd1-Cy5 molecules and the majority of liposomes are empty, and (ii) adding proteoliposomes from a 20:1 mix from unlabeled Hrd1 to the 1:20 mix with Hrd1-Cy5, which should bring back the ubiquitination of the Hrd1-Cy5 monomers.

Major comments:

1. line 135ff, Fig. 2: It would be interesting to assess in how far the number of bleaching steps from liposomes correlates with a Poisson distribution. For this, the fraction of liposomes without protein (accessible via the NBD staining) would be very important, especially for the 1:20 condition where most liposomes are empty. If, as expected from the presented data, the fraction with 2 bleaching steps is much higher than expected from the fraction of empty liposomes, it would strongly support the conjecture that the proteins get integrated into the liposomes as dimers, rather than get integrated as monomers in two separate events.

2. In the ubiquitination assay using fluorescence imaging, it looks like in the condition without ATP, the authors claim that the small signals from ubiquitin-Cy3 that co-localize with Hrd1-Cy5 are from non-specific binding of ubiquitin-Cy3 to proteoliposomes. They should confirm this by repeating the assay without Hrd1-Cy5, and label the proteoliposomes e.g. with NBD. Then they should also see the non-specific binding. If they cannot observe this, I would assume that there is some residual binding of ubiquitin-Cy3 to Hrd1-Cy5 even without ATP. I also disagree with the statement in line 195 that the co-localization of Hrd1-Cy5 with ubiquitin-Cy3 is negligible; I can clearly see many co-localizing spots when overlaying the two images in Fig. 3C (top, left and middle). Many of the brighter ubiquitin-Cy3 spots (probably more than 50%, depending on the threshold) co-localize with a Hrd1-Cy5 spot.

3. In the same context, the authors should quantify the number of ubiquitin-Cy3 and TUBE-Cy3 molecules in a spot by using the step counting. Although they (for unclear reasons) do not use their high-efficiency labeling procedure for ubiquitin or TUBE, they would obtain an estimate for the number of ubiquitins bound to their Hrd1 proteoliposomes. The correlation between the Hrd1-Cy5 and ubiquitin-Cy3 or TUBE-Cy3 molecule numbers would deliver valuable information about the ubiquitination mechanism. The same is true for the correlation between ubiquitin and TUBE, which

could be done using a third color, e.g. in the green range, or by leaving Hrd1 unlabeled and having TUBE or ubiquitin labeled by Cy5. If counting only works with Cy5, they could switch the labels.

4. The last experiment with the added empty liposomes gives a very interesting result; a possible and likely explanation is that at least a dimer is required to induce autoubiquitination. The observation that there is indeed a significant autoubiquitination visible in the proteoliposomes with two bleaching steps suggests that the dimer is also sufficient. However, the authors should follow this lead and test their hypothesis: In the condition with the 1:20 dilution, where the large majority of proteoliposomes carry one or two Hrd1-Cy5 (Fig 2C, right bar), only the proteoliposomes with the 2 bleaching steps should be ubiquitinated. This should even happen without adding the spacer liposomes, since >95% of liposomes are empty. In addition, the proteoliposomes with a single bleaching step should be ubiquitinated when an excess of unlabeled 20:1 proteoliposomes are added. Then the unlabeled Hrd1 should act on the monomeric Hrd1-Cy5 in trans. It is unclear why the authors did not add these obvious but essential experiments, since they require no additional tools but would give very strong support for the conclusions drawn, and therefore strengthen the manuscript.

5. An additional experiment that would support the claim that autoubiquitination of monomers proceeds in trans by other proteoliposomes, or deliver additional information on the mechanisms involved, would be to mix mutant, catalytically inactive Hrd1-Cy5 with wt (unlabeled) Hrd1 and see if the proteoliposomes with one Hrd1-Cy5 bleaching steps show ubiquitin-Cy3 signals.

6. Methods: (i) The origin of Sortase A is unclear. Was it purchased or expressed and purified by the authors? (ii) If linkers were used between Hrd1, the sortase tag and/or the SBP epitope tag, their aa sequence should be stated. (iii) When special substances are used, product numbers from the companies should be given in addition to the manufacturer's name, since sometimes there are different types (e.g. Biotin-PEG-MVA is available in several MW versions from Laysan).

Minor comments:

line 40: Do the authors mean cytoplasm or cytosol?

Fig. 1b, c: The lack of a signal in Fig. 1c "Bottom" suggests that there is no unincorporated protein, yet the sketch in Fig. 1b and the figure legend for 1b suggest that there is a significant amount of it. Maybe the authors could clarify if either the unincorporated protein is not visible (or somewhere else), or if basically all of the protein was incorporated into the proteoliposomes, leading to the virtual absence of a signal.

line 118, Suppl. Fig. 1: There is some mismatch between the main text and the suppl. figure, and the suppl. fig. legend. (i) C and D labels in the figure seem to be switched compared to the legend. (ii) the text talks about the relation between Cy5 intensity and number of fluorophores, but there is sth else depicted in Suppl. Fig. 1c. (iii) In the text, the authors claim that "Hrd1-Cy5 largely colocalized with NBD-PC", but in Suppl. Fig 1, the pie chart shows 37%.

line 124: It is unclear what the authors mean by "to define minimal stoichiometries within proteoliposomes".

line 131: A lower limit for what? I assume the authors mean that they want to dilute the Hrd1-Cy5 in the sample as much as possible to reduce the probability for getting two protein complexes within the same proteoliposome. But at the same time, diluting means getting a smaller number of observations, i.e. protein containing liposomes, in the microscopy. Therefore, they probably are looking for the lowest amount/highest dilution of Hrd1-Cy5 that still gives them an acceptable density of spots in the microscopy. But this is rather unclear and should be better explained.

Suppl. Fig. 2a: Where is the 20:1 condition that is mentioned in the main text and shown in Fig. 2a? An interpretation for the observations would be preferable, e.g. if the results are expected, or if/how the differences/similarities might have an impact of the experiment.

line 141f: "The Cy5 fluorescence intensity of an individual diffraction limited foci is proportional to the number of Hrd1-Cy5 molecules within the spot." This is not true since the proteoliposomes experience different excitation intensities. As visible in Fig. 2B, the step sizes are 200 units in some spots, but 500 in others. This is often caused by an uneven illumination profile, e.g. from lack of correction for the laser's Gaussian profile, interference fringes of the laser, etc.

line 147: I assume that also with a lower labeling efficiency, the step counting itself would be reliable. It is rather the assignment of protein molecule number from the step count that becomes less reliable.

line 121, 152: By "oligomers", do the authors refer to a protein complex containing multiple protein molecules, or that multiple proteins are in one proteoliposome? In the latter case, I would not call that an "oligomer". Same in line 162, where they could write call this "spots with higher number of bleaching events" or a similar term, and line 205 and 340/341, "oligomeric classes".

Fig. 2d, Suppl. Fig. 2b: I find "largest possible oligomer" a bit misleading since the authors simulate a pure population of the respective multimer.

line 159, 341: The authors should check if the term "Poisson limit" is really what they mean. As far as I remember (and read), this refers to the similarity of Poisson and binomial distributions for certain conditions, whereas I believe the authors mean the case where the incorporation probability is very low, such that $\Pr(X=1) \gg \Pr(X=2)$.

line 160: "discrete" would be true also for two or three insertion events. Maybe "single" or "one at most" is better.

line 166ff: the authors should better discriminate between "monomers" that refer to a monomeric protein, and "monomers" that are spots with a single bleaching step. I recommend to not use the "monomer" term for both since it causes confusion (same for "dimers").

line 173: "unlikely": no matter how close the labeling efficiency is to 100%, as long as it is below 100%, the expected distribution of spot counts will be left-shifted compared to the distribution of multimerization states. I would rather say that with a high labeling efficiency, the shift can be neglected.

line 211, Fig. 3h: "TUBE": clarification that this protein binds to up to 6 ubiquitins (as stated in the methods section) would be good, and this should be consistent with the sketch.

line 305: it is not entirely clear what "This" refers to.

Reviewer #2 (Remarks to the Author):

The manuscript submitted to Nature Communications outlines a technical tour-de-force from the Baldrige lab, describing two assays that ultimately allow the investigators to conclude that dimeric and higher order Hrd1 oligomers represent the ubiquitination-active form of the protein. The importance of this result is underscored by the need for refined assays amongst the complex activities exhibited by Hrd1—as a retrotranslocon, lipid-thinning protein, and ubiquitin ligase—and by structural studies that have variably positioned the protein as the de facto retrotranslocon or a component of the retrotranslocon, along with Der1. Yet, as noted by the authors, Hrd1 overexpression is sufficient to support retrotranslocation/ERAD in the absence of its partners, and recent data have providing evidence for channel/ion translocation activity. Also noted by the authors, the methods reported in this well written manuscript will be applicable to those studying analogous systems and surmount secondary effects from crosslinking and assays in detergent micelles. The system will additionally prove invaluable to allow for add-back reconstitutions with other (e.g., Der1) components of the ERAD complex with which Hrd1 might associate once ubiquitinated (as noted in the Discussion section).

The first assay employs a Hrd1-sortase peptide-SBP fusion to which Cy5 could be added at near stoichiometric levels. Along with removal of the SBP tag and reconstitution into proteoliposomes (containing a fluorescent reporter), TIRF microscopy and photobleaching allowed these investigators to count the number of Hrd1 molecules in each of the many existing complexes. The second assay—which allowed for conjugation with Cy3-labeled ubiquitin and examinations of co-localization—positioned Assinaur and colleagues to determine that Hrd1 dimers and higher order complexes reflect the functional unit for K-48 ubiquitination activity, a conclusion that was made possible by titrating Hrd1:liposome ratios and incubation with a tandem ubiquitin binding element (TUBE) that was also Cy3 labeled. A subsequent titration of control liposomes lacking Hrd1 built confidence that ubiquitination was independent of inter-proteoliposome interactions, although this might have explained some of the residual monomer activity.

Overall, this study is clearly described and will be welcome by investigators in the ERAD research

community as well as those who wish to study the dynamic interactions exhibited by membrane protein complexes. Relatively minor suggestions are made to strengthen an already formidable study.

Comments:

1. One concern surrounds the sometimes emphatic statements that the monomer is inactive, or that "monomers are incapable of autoubiquitination (Figure 4B)". If fact, there appears to be residual activity that could be ascribed to the monomer in this figure and elsewhere. Unless data to the contrary are conclusive, the authors are encouraged to soften the text where appropriate, i.e., that the data strongly suggest that the dimer represents the minimally active species.
2. With regard to this fact, a formal possibility exists that Hrd1 monomers are more labile after reconstitution into proteoliposomes, whereas dimers and multimers are stabilized and thus retain ubiquitin ligase activity. Do the authors have any evidence to the contrary?
3. Although this concern is likely irrelevant for the ubiquitination assay, is there any notion of how many Hrd1-Cy3 derivatives were incorporated into proteoliposomes in an inverse orientation?
4. Given the long incubation time of the ubiquitination assay (60 min), the authors note the possibility that ubiquitination changes the stoichiometry of dimers and higher order oligomers in the bilayer, at least within this time frame. Do the data change during a time course? Such an experiment could yield these data, although it is also possible that the signal is too low to perform this analysis at early time points. However, one might imagine that dynamic state changes could be used to create a ubiquitination pool of Hrd1 in the ER during certain cell states.
5. The legend in figure 3A states that "empty liposomes" were used as a control, but the figure appears to instead show a -ATP control.

Reviewer #1 (Remarks to the Author):

In the manuscript "Direct observation of autoubiquitination for an integral membrane ubiquitin ligase in ERAD" by Assainar et al., the authors investigate the minimal multimerization state required for the ubiquitin ligase Hrd1 to be active in the sense that it causes its autoubiquitination. They develop a single-molecule imaging assay based on proteoliposomes. After purification and labeling of Hrd1 with Cy5, the protein gets integrated into the liposomes. Importantly, highly efficient labeling (>95%) was achieved with a newly developed sortase-based approach. The number of Hrd1 molecules integrated into individual liposomes was determined with a subunit-counting method based on Cy5 photobleaching steps. For low amounts of protein, the authors see mainly one or two proteins per proteoliposome. Next, they used Cy3-labeled ubiquitin to test if the proteins are able to induce autoubiquitination. First, they observe that a majority of all proteoliposomes get ubiquitinated. However, when they add empty liposomes as spacers, the proteoliposomes displaying a single photobleaching step do not show ubiquitination, but all the others still do. Their conclusion is that at least two Hrd1 molecules in the same liposome are required for autoubiquitination.

I think that the study is very well done and shows a novel combination of approaches for single-molecule analysis. The observation that the proteoliposomes with a single photobleaching step do not show ubiquitination after dilution with the spacer liposomes is very exciting. However, at this point, the authors fail to test their two hypotheses that the underlying cause is indeed that monomers are unable to lead to autoubiquitination, and that the lack of spacer liposomes allow trans-ubiquitination of the monomers by active complexes in other proteoliposomes. I am quite surprised that the authors did not do these experiments since (in my view) no additional tools or developments would be needed.

Therefore, in the current state, I would consider this work as somehow incomplete and recommend a revision of the manuscript to allow addition of the experiments as suggested in major comment 4: (i) determining the fraction of ubiquitinated proteoliposomes in the 1:20 dilution, where proteoliposomes contain only 1 or 2 Hrd1-Cy5 molecules and the majority of liposomes are empty, and (ii) adding proteoliposomes from a 20:1 mix from unlabeled Hrd1 to the 1:20 mix with Hrd1-Cy5, which should bring back the ubiquitination of the Hrd1-Cy5 monomers.

Major comments:

1. line 135ff, Fig. 2: It would be interesting to assess in how far the number of bleaching steps from liposomes correlates with a Poisson distribution. For this, the fraction of liposomes without protein (accessible via the NBD staining) would be very important, especially for the 1:20 condition where most liposomes are empty. If, as expected from the presented data, the fraction with 2 bleaching steps is much higher than expected from the fraction of empty liposomes, it would strongly support the conjecture that the proteins get integrated into the liposomes as dimers, rather than get integrated as monomers in two separate events.

Thank you for the suggestion to include an experimental validation of the titration experiments. We performed this experiment and included the results as a new supplemental figure (Supplementary Figure 2b). Briefly, we reconstituted Hrd1 liposomes using fluorescent

NBD lipids at 20:1 and 1:20 ratios, immobilized the liposomes on a glass surface, and imaged individual Hrd1 containing proteoliposomes. Our experiment demonstrates that at the highest protein density reconstitutions (20:1), we noted that 10% of proteoliposomes contain at least one Hrd1 molecule. We observed the opposite result when performing reconstitutions under low protein density (1:20) in which case the vast majority of liposomes are empty with as few as 1% of proteoliposomes containing at least one Hrd1 molecule. To identify sufficient events, we normally immobilize at a surface density of at least 100 Hrd1 proteoliposomes per field of view, but in this case we needed to cover our surface with a density of liposomes that is so high that it makes it impossible to resolve individual molecules. This measurement demonstrates that we have reached the Poisson limit, a condition where most liposomes contain no protein and some liposomes contain at least one Hrd1 molecule.

2. In the ubiquitination assay using fluorescence imaging, it looks like in the condition without ATP, the authors claim that the small signals from ubiquitin-Cy3 that co-localize with Hrd1-Cy5 are from non-specific binding of ubiquitin-Cy3 to proteoliposomes. They should confirm this by repeating the assay without Hrd1-Cy5, and label the proteoliposomes e.g. with NBD. Then they should also see the non-specific binding. If they cannot observe this, I would assume that there is some residual binding of ubiquitin-Cy3 to Hrd1-Cy5 even without ATP. I also disagree with the statement in line 195 that the co-localization of Hrd1-Cy5 with ubiquitin-Cy3 is negligible; I can clearly see many co-localizing spots when overlaying the two images in Fig. 3C (top, left and middle). Many of the brighter ubiquitin-Cy3 spots (probably more than 50%, depending on the threshold) co-localize with a Hrd1-Cy5 spot.

Thank you for bringing this to our attention. Even though the colocalization of ubiquitin^{Cy3} with Hrd1^{Cy5} was low in the absence of ATP, we agree that it was important to test whether the ubiquitin^{Cy3} could be sticking to either the liposomes or the Hrd1^{Cy5}. To test this possibility, we added a new experiment with empty liposomes to our manuscript. These data are included as a new figure (Supplementary Fig. 3b). We noted a finite but small fraction of empty liposomes with ubiquitin^{Cy3} non-specifically bound and altered the text to reflect the reviewer's suggestion.

3. In the same context, the authors should quantify the number of ubiquitin-Cy3 and TUBE-Cy3 molecules in a spot by using the step counting. Although they (for unclear reasons) do not use their high-efficiency labeling procedure for ubiquitin or TUBE, they would obtain an estimate for the number of ubiquitins bound to their Hrd1 proteoliposomes. The correlation between the Hrd1-Cy5 and ubiquitin-Cy3 or TUBE-Cy3 molecule numbers would deliver valuable information about the ubiquitination mechanism. The same is true for the correlation between ubiquitin and TUBE, which could be done using a third color, e.g. in the green range, or by leaving Hrd1 unlabeled and having TUBE or ubiquitin labeled by Cy5. If counting only works with Cy5, they could switch the labels.

Thank you for the suggestion. We agree that it would be interesting to know the relationship between the number of ubiquitin proteins and the number of TUBE proteins attached to Hrd1. However, there are technical limitations to executing the experiments as suggested. We are unable to label both ubiquitin and sortase using the high-efficiency sortase approach, for the following reasons:

In the case of ubiquitin, the C-terminus is required for conjugation to target proteins, so sortase-labeling would prevent ubiquitin function. For the TUBE protein, the individual ubiquitin binding elements are connected by Gly-Gly-Gly-Gly-Ser-Gly-Gly-Gly linkers. The sortase enzyme will use these sequences as substrates and essentially fragment this protein into individual ubiquitin binding domains, without the ability to recognize ubiquitin chains.

Because this experiment is important, we developed an alternative approach to test the important idea of whether the direct (Ub^{Cy3}) and indirect experiments (TUBE^{Cy3}) yield similar results. We reconstituted unlabeled Hrd1 protein into proteoliposomes and performed ubiquitination experiments using ubiquitin^{Cy5}. Then, we incubated the polyubiquitinated Hrd1 proteoliposomes with TUBE^{Cy3} and immobilized the proteoliposomes for imaging. The results of this experiment are provided as a new figure (Supplementary Fig. 3e). Our data confirms that the direct and indirect ubiquitination experiments are roughly correlated, with the direct ubiquitination experiments being more sensitive than orthogonal detection with TUBE.

4. The last experiment with the added empty liposomes gives a very interesting result; a possible and likely explanation is that at least a dimer is required to induce autoubiquitination. The observation that there is indeed a significant autoubiquitination visible in the proteoliposomes with two bleaching steps suggests that the dimer is also sufficient. However, the authors should follow this lead and test their hypothesis: In the condition with the 1:20 dilution, where the large majority of proteoliposomes carry one or two Hrd1-Cy5 (Fig 2C, right bar), only the proteoliposomes with the 2 bleaching steps should be ubiquitinated. This should even happen without adding the spacer liposomes, since >95% of liposomes are empty. In addition, the proteoliposomes with a single bleaching step should be ubiquitinated when an excess of unlabeled 20:1 proteoliposomes are added. Then the unlabeled Hrd1 should act on the monomeric Hrd1-Cy5 in trans. It is unclear why the authors did not add these obvious but essential experiments, since they require no additional tools but would give very strong support for the conclusions drawn, and therefore strengthen the manuscript.

Thank you for this question! We did attempt the experiments the reviewer proposed using the 1:20 proteoliposomes, before the original submission of the manuscript, but ran into technical issues. We rarely observed ubiquitination under these very dilute conditions. This was true even for proteoliposomes with larger (>4) numbers of Hrd1 in which case we easily observe the assembly of polyubiquitin chains under our 20:1 reconstitution conditions.

We tried to mix unlabeled Hrd1 proteoliposomes at 20:1 with Hrd1^{Cy5} reconstituted at 1:20, to see if the ubiquitination rate goes up. However we did not see any ubiquitination in this condition either. This experiment is at the limit of detection of our current experimental setup and is effectively adding a set of empty liposomes (due to the extreme difference in protein occupancies in these reconstitution conditions as noted in new Supplementary Fig. 2b). We believe that the Hrd1 concentrations in these reactions were ultimately too low (initial reconstitution conditions started at ~4nM, final total Hrd1 concentration below 1nM). We don't have accurate estimates for affinities between Hrd1 and Ubc7 (the E2) but for most E2-RING interactions the K_d are in the uM range. Although, these enzymes generally function well below the K_d of interaction, we were unable to detect ubiquitination under these very dilute conditions.

Nevertheless, we started developing a new experimental setup to ultimately address the reviewer's question. We started by immobilizing our proteoliposomes on the slide surface to prevent interactions between Hrd1 in *trans*. Then we washed away the proteoliposomes that were not surface immobilized and flowed in ubiquitination mix, including ATP, onto the slide surface and incubated for 60 minutes. Then we washed the slide surface extensively, and added 10pM labeled TUBE^{Cy3} incubated for 10 minutes and then collected our movies. Our preliminary results showed that proteoliposomes with larger (>4) numbers of Hrd1 efficiently assembled polyubiquitin chains, dimers were less efficient but well above background, whereas monomers failed to do so.

We've attached the results for the reviewers, because we believe these data support our conclusions. However, these experiments are suboptimal at the moment. We think that pushing this new method forward will require a significant investment of time and resources which is beyond the scope of this manuscript.

We also believe that the new time course data in Supplementary Fig. 4a provides additional support for our conclusions. At shorter time periods, we see very little ubiquitination of Hrd1 monomers, and dimers, but in the later time periods, dimers are also becoming ubiquitinated.

5. An additional experiment that would support the claim that autoubiquitination of monomers proceeds in *trans* by other proteoliposomes, or deliver additional information on the mechanisms involved, would be to mix mutant, catalytically inactive Hrd1-Cy5 with wt (unlabeled) Hrd1 and see if the proteoliposomes with one Hrd1-Cy5 bleaching steps show ubiquitin-Cy3 signals.

This is an interesting idea. Previous studies have demonstrated that wild-type Hrd1 does not ubiquitinate Hrd1(C399S) in detergent micelles (<https://doi.org/10.1016/j.cell.2014.07.050>). The reason for this observation is unclear. However, because we have developed a new, potentially more sensitive assay, we wanted to test whether this was true even with the *trans* ubiquitination reaction we observed in proteoliposomes. We purified Hrd1(C399S), labeled it with Cy5, and reconstituted it at 20:1 ratio in biotin-lipid containing proteoliposomes. We mixed these proteoliposomes with wild-type Hrd1 reconstituted (also 20:1, but no biotin), performed the ubiquitination assay, and immobilized only the Hrd1(C399S) liposomes on the slide surface. We

observed very little colocalization of Hrd1(C399S) with ubiquitin^{Cy3} and these data have been included as new figures (Supplementary Fig. 4c,d). These experiments provide important context, and demonstrate the specificity of our reaction conditions. However, because Hrd1(C399S) is not ubiquitinated by wild-type Hrd1 these data don't allow us to unambiguously address this question.

6. Methods: (i) The origin of Sortase A is unclear. Was it purchased or expressed and purified by the authors? (ii) If linkers were used between Hrd1, the sortase tag and/or the SBP epitope tag, their aa sequence should be stated. (iii) When special substances are used, product numbers from the companies should be given in addition to the manufacturer's name, since sometimes there are different types (e.g. Biotin-PEG-MVA is available in several MW versions from Laysan).

We apologize for this oversight and have included the information the reviewer suggests in the methods section (lines 396, 468, 562). Yes, we expressed and purified the sortase A (pentamutant) developed by the Liu lab (<https://doi.org/10.1073/pnas.110104610>).

Minor comments:

line 40: Do the authors mean cytoplasm or cytosol?

Thank you for catching the error, we've updated the text.

Fig. 1b, c: The lack of a signal in Fig. 1c "Bottom" suggests that there is no unincorporated protein, yet the sketch in Fig. 1b and the figure legend for 1b suggest that there is a significant amount of it. Maybe the authors could clarify if either the unincorporated protein is not visible (or somewhere else), or if basically all of the protein was incorporated into the proteoliposomes, leading to the virtual absence of a signal.

Thank you for bringing this to our attention. As the reviewer correctly inferred, there is no unincorporated protein left behind at the bottom of the gradient (Fig. 1c). Generally including flotation in our work flow is a fail-safe measure since most of the unincorporated protein is cleared out in earlier steps. The model in Fig. 1b is supposed to illustrate the process, so we updated Fig. 1b to match with what is observed in the experiment.

line 118, Suppl. Fig. 1: There is some mismatch between the main text and the suppl. figure, and the suppl. fig. legend. (i) C and D labels in the figure seem to be switched compared to the legend. (ii) the text talks about the relation between Cy5 intensity and number of fluorophores, but there is sth else depicted in Suppl. Fig. 1c. (iii) In the text, the authors claim that "Hrd1-Cy5 largely colocalized with NBD-PC", but in Suppl. Fig 1, the pie chart shows 37%.

We apologize for the oversight. (i) we made appropriate changes to the figure legends. (ii) Based on our quantification of Fig. 1f (depicted in Supplementary Fig. 1d), 37% of the total spots had Hrd1^{Cy5} colocalized with NBD-PC, whereas 10% of the spots had Hrd1^{Cy5} that did not colocalize. We made the claim that Hrd1^{Cy5} is mostly colocalized with NBD-PC, which is even more apparent in the new Supplementary Fig. 2b.

We would like to note that 53% of the liposomes are empty in this experiment (Fig. 1f and Supplementary Fig. 1d). In the new Supplementary Fig. 2b, we found about 90% of liposomes were empty, and all of the Hrd1 was colocalized with NBD-PC.

line 124: It is unclear what the authors mean by "to define minimal stoichiometries within proteoliposomes".

We have revised the text to improve clarity.

line 131: A lower limit for what? I assume the authors mean that they want to dilute the Hrd1-Cy5 in the sample as much as possible to reduce the probability for getting two protein complexes within the same proteoliposome. But at the same time, diluting means getting a smaller number of observations, i.e. protein containing liposomes, in the microscopy. Therefore, they probably are looking for the lowest amount/highest dilution of Hrd1-Cy5 that still gives them an acceptable density of spots in the microscopy. But this is rather unclear and should be better explained.

We have revised the text to improve the clarity.

Suppl. Fig. 2a: Where is the 20:1 condition that is mentioned in the main text and shown in Fig. 2a? An interpretation for the observations would be preferable, e.g. if the results are expected, or if/how the differences/similarities might have an impact of the experiment.

Unfortunately, we did not measure the fluorescence intensity using a fluorescence plate reader for every reconstitution condition we tested. We only did so for three conditions depicted in Supplementary Fig. 2a as validations of our reconstitution approach. As a general practice, we visualize the fluorescence of each gradient fraction in PCR strip tubes using a fluorescence scanner as depicted in Fig. 1d. As suggested, we added a sentence to support our interpretation (line 129).

line 141f: "The Cy5 fluorescence intensity of an individual diffraction limited foci is proportional to the number of Hrd1-Cy5 molecules within the spot." This is not true since the proteoliposomes experience different excitation intensities. As visible in Fig. 2B, the step sizes are 200 units in some spots, but 500 in others. This is often caused by an uneven illumination profile, e.g. from lack of correction for the laser's Gaussian profile, interference fringes of the laser, etc.

Thanks for bringing this up, we are aware of the uneven illumination profile which is a common issue with imaging single molecules using an objective TIRF configuration. Importantly, we did not use absolute intensity changes in our analysis to estimate protein numbers within a particular liposome. Our goal was to provide an easy guide to readers unfamiliar with single-molecule microscopy to be able to interpret our data. However, we agree that we need to make this distinction in the text and updated this section in the manuscript to reflect the reviewer's input.

line 147: I assume that also with a lower labeling efficiency, the step counting itself would be reliable. It is rather the assignment of protein molecule number from the step count that becomes less reliable.

Thanks for the suggestion. We modified the text to include the reviewer's suggestion.

line 121, 152: By "oligomers", do the authors refer to a protein complex containing multiple protein molecules, or that multiple proteins are in one proteoliposome? In the latter case, I would not call that an "oligomer". Same in line 162, where they could write call this "spots with higher number of bleaching events" or a similar term, and line 205 and 340/341, "oligomeric classes".

We appreciate the reviewer's thought process and have made changes in most of the mentioned locations. However, in certain places, we feel it is appropriate to use this terminology (oligomers) since we observed 1,2,3,4 and >4 Hrd1 containing liposomes at the two lowest protein concentrations that we tested in our reconstitution assay. Although we cannot make inferences about the exact oligomeric state of Hrd1 based on step distributions at the higher protein concentrations, our titration covers a wide range of lipid:protein ratios which strongly supports our observations.

Fig. 2d, Suppl. Fig. 2b: I find "largest possible oligomer" a bit misleading since the authors simulate a pure population of the respective multimer.

Thank you for the suggestion. We replaced "largest possible oligomer" with "simulated step distribution of a population of n-mers"

line 159, 341: The authors should check if the term "Poisson limit" is really what they mean. As far as I remember (and read), this refers to the similarity of Poisson and binomial distributions for certain conditions, whereas I believe the authors mean the case where the incorporation probability is very low, such that $\Pr(X=1) \gg \Pr(X=2)$.

line 160: "discrete" would be true also for two or three insertion events. Maybe "single" or "one at most" is better.

Thanks for bringing these important distinctions to our attention. We have incorporated these changes to our manuscript.

line 166ff: the authors should better discriminate between "monomers" that refer to a monomeric protein, and "monomers" that are spots with a single bleaching step. I recommend to not use the "monomer" term for both since it causes confusion (same for "dimers").

Thanks for the suggestion. We modified the text to include the reviewer's suggestion.

line 173: "unlikely": no matter how close the labeling efficiency is to 100%, as long as it is below 100%, the expected distribution of spot counts will be left-shifted compared to the distribution of multimerization states. I would rather say that with a high labeling efficiency, the shift can be neglected.

Thanks for the suggestion. We modified the text to include the reviewer's suggestion.

line 211, Fig. 3h: "TUBE": clarification that this protein binds to up to 6 ubiquitins (as stated in the methods section) would be good, and this should be consistent with the sketch.

Thanks for the suggestion. We modified the text and figure to include the reviewer's suggestion.

Reviewer #2 (Remarks to the Author):

The manuscript submitted to Nature Communications outlines a technical tour-de-force from the Baldrige lab, describing two assays that ultimately allow the investigators to conclude that dimeric and higher order Hrd1 oligomers represent the ubiquitination-active form of the protein. The importance of this result is underscored by the need for refined assays amongst the complex activities exhibited by Hrd1—as a retrotranslocon, lipid-thinning protein, and ubiquitin ligase—and by structural studies that have variably positioned the protein as the de facto retrotranslocon or a component of the retrotranslocon, along with Der1. Yet, as noted by the authors, Hrd1 overexpression is sufficient to support retrotranslocation/ERAD in the absence of its partners, and recent data have providing evidence for channel/ion translocation activity. Also noted by the authors, the methods reported in this well written manuscript will be applicable to those studying analogous systems and surmount secondary effects from crosslinking and assays in detergent micelles. The system will additionally prove invaluable to allow for add-back reconstitutions with other (e.g., Der1) components of the ERAD complex with which Hrd1 might associate once ubiquitinated (as noted in the Discussion section).

The first assay employs a Hrd1-sortase peptide-SBP fusion to which Cy5 could be added at near stoichiometric levels. Along with removal of the SBP tag and reconstitution into proteoliposomes (containing a fluorescent reporter), TIRF microscopy and photobleaching allowed these investigators to count the number of Hrd1 molecules in each of the many existing complexes. The second assay—which allowed for conjugation with Cy3-labeled ubiquitin and examinations of co-localization—positioned Assinaur and colleagues to determine that Hrd1 dimers and higher order complexes reflect the functional unit for K-48 ubiquitination activity, a conclusion that was made possible by titrating Hrd1:liposome ratios and incubation with a tandem ubiquitin binding element (TUBE) that was also Cy3 labeled. A subsequent titration of control liposomes lacking Hrd1 built confidence that ubiquitination was independent of inter-proteoliposome interactions, although this might have explained some of the residual monomer activity.

Overall, this study is clearly described and will be welcome by investigators in the ERAD research community as well as those who wish to study the dynamic interactions exhibited by membrane protein complexes. Relatively minor suggestions are made to strengthen an already formidable study.

Comments:

1. One concern surrounds the sometimes emphatic statements that the monomer is inactive, or that “monomers are incapable of autoubiquitination (Figure 4B)”. If fact, there appears to be residual activity that could be ascribed to the monomer in this figure and elsewhere. Unless data to the contrary are conclusive, the authors are encouraged to soften the text where appropriate, i.e., that the data strongly suggest that the dimer represents the minimally active species.

Our data provides strong support for a model in which a Hrd1 dimer is minimally required for efficient auto ubiquitination. We agree with the reviewer that this model does not dismiss the possibility of the monomer having residual or minimal activity. We have made changes to the text to soften our conclusions.

2. With regard to this fact, a formal possibility exists that Hrd1 monomers are more labile after reconstitution into proteoliposomes, whereas dimers and multimers are stabilized and thus retain ubiquitin ligase activity. Do the authors have any evidence to the contrary?

We are unsure exactly what the reviewer means by “labile”, but we have interpreted this comment to mean that the monomer is more likely to be inactive compared to other larger Hrd1 assemblies. We do not have strong evidence that the monomers are more prone to inactivation. However, under reaction conditions where Hrd1 was reconstituted using high ratios of protein to lipid, we can clearly see autoubiquitination of Hrd1 monomers. This data is further supported by our experiments using Hrd1(C399S) (Supplementary Fig. 4c) where we were unable to observe any ubiquitination either in *cis* or in *trans*. This means we require “active” Hrd1 on both liposomes for the *trans* reaction to work. Hence, we think that monomers being more labile compared to other oligomeric species is less likely.

Still, we agree we cannot rule out this possibility and have updated the discussion to reflect this (line 324).

3. Although this concern is likely irrelevant for the ubiquitination assay, is there any notion of how many Hrd1-Cy3 derivatives were incorporated into proteoliposomes in an inverse orientation.

To test the orientation, we used the sortase A enzyme to test for accessibility of the Hrd1 C-terminus and included the results in a new figure (Supplementary Fig. 1f). Briefly, Hrd1^{Cy5} proteoliposomes were incubated with unlabeled GGGC peptide, sortase A, with or without detergent solubilization. We found that the C-terminus of most Hrd1 proteins were accessible (>85%, with 94% after detergent solubilization) meaning Hrd1 was mostly inserted with its cytosolic side facing outward in the liposome.

4. Given the long incubation time of the ubiquitination assay (60 min), the authors note the possibility that ubiquitination changes the stoichiometry of dimers and higher order oligomers in the bilayer, at least within this time frame. Do the data change during a time course? Such an experiment could yield these data, although it is also possible that the signal is too low to perform this analysis at early time points. However, one might imagine that dynamic state changes could be used to create a ubiquitination pool of Hrd1 in the ER during certain cell states.

We thank the reviewer for this important question. We tested the ubiquitination status across different Hrd1 stoichiometries at different time points (15 min, 30 min and 60 min). We noticed that large (>4) oligomers of Hrd1 underwent autoubiquitination much sooner than dimers (or monomers). We are not able to ascribe ubiquitination rates to particular Hrd1 configurations within proteoliposomes containing larger amounts of Hrd1. We also cannot discern whether Hrd1 stoichiometry changes within liposomes because of the diffraction limited nature of our microscopy approach i.e., we cannot tell if Hrd1 oligomers are dispersed or clustered within a

single proteoliposome. This time-course data has now been added to the manuscript as a new figure (Supplementary Fig. 4a).

5. The legend in figure 3A states that “empty liposomes” were used as a control, but the figure appears to instead show a -ATP control

Thank you for catching this. We apologize for the mistake in the legend and have fixed it.

REVIEWERS' COMMENTS

Reviewer #2 (Remarks to the Author):

The authors have satisfactorily addressed my previous concerns via the addition of text and two new figure panels. With regard to the time course, is the increase in dimer ubiquitination over time statistically significant? If not, then the description of this experiment in the main section of the manuscript might be more explicit about the possibility that higher order oligomers are the most active species--perhaps due to increased avidity or reaction efficiency?--yet perhaps these species break down during the course of the experiment, giving rise to the slower and less intense increase in dimer ubiquitination.

Reviewer #2 (Remarks to the Author):

The authors have satisfactorily addressed my previous concerns via the addition of text and two new figure panels. With regard to the time course, is the increase in dimer ubiquitination over time statistically significant? If not, then the description of this experiment in the main section of the manuscript might be more explicit about the possibility that higher order oligomers are the most active species--perhaps due to increased avidity or reaction efficiency?--yet perhaps these species break down during the course of the experiment, giving rise to the slower and less intense increase in dimer ubiquitination.

We thank the reviewer for this important point. The increase in dimer over time is not statistically significant, so we updated the text to be more explicit to reflect the reviewer's point.

At this point, we can't be completely sure whether this is because of higher avidity of larger complexes of Hrd1 or if it is due to greater Hrd1 numbers being present in these particular liposomes (meaning more sites for ubiquitination). However, since the large complexes showed ubiquitination at earlier time points, it might speak to the avidity of the species. But ,we don't think this evidence alone is enough to make a strong conclusion, so we have elected not to focus on this point until we have better experiments with more conclusive data.